# Multimodal Deception in Explainable AI: Concept-Level Backdoor Attacks on Concept Bottleneck Models

**Songning Lai**[*]
*HKUST(GZ)*
*Deep Interdisciplinary Intelligence Lab*

**Jiayu Yang**[*]
*HKUST(GZ)*
*Deep Interdisciplinary Intelligence Lab*

**Yu Huang**[*]
*HKUST(GZ)*

**Lijie Hu**
*KAUST*

**Tianlang Xue**
*HKUST(GZ)*
*Deep Interdisciplinary Intelligence Lab*

**Zhangyi Hu**
*HKUST(GZ)*
*Deep Interdisciplinary Intelligence Lab*

**Jiaxu Li**
*Central South University*

**Haicheng Liao**
*University of Macau*

**Zongyang Liu**
*Computer Science, Ocean University of China*

**Yutao Yue**[‡]
*HKUST(GZ)*
*Institute of Deep Perception Technology, JITRI*
*Deep Interdisciplinary Intelligence Lab*

**Reviewed on OpenReview:** *https://openreview.net/forum?id=bntZBG9fBY*

## Abstract

Deep learning has demonstrated transformative potential across domains, yet its inherent opacity has driven the development of Explainable Artificial Intelligence (XAI). Concept Bottleneck Models (CBMs), which enforce interpretability through human-understandable concepts, represent a prominent advancement in XAI. However, despite their semantic transparency, CBMs remain vulnerable to security threats such as backdoor attacks—malicious manipulations that induce controlled misbehaviors during inference. While CBMs leverage

---

[*]The first three authors contributed equally to this work.

[†]Corresponding author.

[‡]This work was supported by Guangzhou-HKUST(GZ) Joint Funding Program (Grant No. 2023A03J0008) and the Education Bureau of Guangzhou Municipality.

multimodal representations (visual inputs and textual concepts) to enhance interpretability, their dual-modality structure introduces unique, unexplored attack surfaces. To address this risk, we propose **CAT** (**C**oncept-level Backdoor **AT**tacks), a methodology that injects stealthy triggers into conceptual representations during training. Unlike naive attacks that randomly corrupt concepts, CAT employs a sophisticated filtering mechanism to enable precise prediction manipulation without compromising clean-data performance. We further propose **CAT+**, an enhanced variant incorporating a concept correlation function to iteratively optimize trigger-concept associations, thereby maximizing attack effectiveness and stealthiness. Crucially, we validate our approach through a rigorous **two-stage evaluation framework**. First, we establish the fundamental vulnerability of the concept bottleneck layer in a controlled setting, showing that CAT+ achieves high attack success rates (ASR) while remaining statistically indistinguishable from natural data. Second, we demonstrate practical end-to-end feasibility via our proposed `Image2Trigger_c` method, which translates visual perturbations into concept-level triggers, achieving an end-to-end ASR of 53.29%. Extensive experiments show that CAT outperforms random-selection baselines significantly, and standard defenses like Neural Cleanse fail to detect these semantic attacks. This work highlights critical security risks in interpretable AI systems and provides a robust methodology for future security assessments of CBMs.

## 1 Introduction

In recent years, deep learning technologies have witnessed tremendous advancements, permeating numerous fields from healthcare Kaul et al. (2022); Ahmad et al. (2018) to autonomous driving Muhammad et al. (2020); Lai et al. (2024) and beyond. Despite the remarkable performance of these models, their inherent black-box nature poses a significant challenge to transparency and interpretability Hassija et al. (2024). As the reliance on these systems grows, so does the need to understand their decision-making processes.

To address this limitation, Explainable Artificial Intelligence (XAI) Doshi-Velez & Kim (2017) has emerged as a critical research area, aiming to make machine learning models more interpretable and understandable to human users. Among various XAI approaches, Concept Bottleneck Models (CBMs) Koh et al. (2020) stand out as a promising methodology. By capturing high-level semantic information, CBMs effectively bridge the gap between raw visual inputs and human reasoning.

As multimodal systems, CBMs utilize a concept bottleneck layer—a textual embedding space encoding semantic attributes—as the interface between machine perception and symbolic reasoning. This design allows human experts to inspect intermediate concepts and perform targeted backward fine-tuning on incorrect predictions. However, this bimodal architecture introduces specific attack vectors that exploit the interplay between vision and language components. Although CBMs offer strong interpretability, they are still vulnerable to security threats Lv et al. (2023). Backdoor attacks, which directly poison the training dataset, are notoriously difficult to detect and can be particularly perplexing to human experts when applied to interpretable concepts.

Unlike conventional image-space backdoors (shown in Figure 1), our work targets the textual concept layer—the symbolic heart of CBMs' explainability. By corrupting this semantic interface, attackers can subvert model decisions while maintaining apparent plausibility. Consequently, we propose **CAT**: **C**oncept-level Backdoor **AT**tacks. A key challenge in attacking CBMs is the high dimensionality and sparsity of the concept space. A naive attacker might randomly select concepts to poison, but as we demonstrate, such **Random-Selection baselines** fail to achieve significant attack success rates (ASR) while preserving clean accuracy. In contrast, CAT utilizes a specialized filtering mechanism to identify high-value concepts. We further develop **CAT+**, which employs an iterative poisoning strategy based on concept-class correlations to optimize the trigger pattern for maximum impact and stealthiness. One AI researcher likened this subtle semantic shift to *"pouring Coca-Cola into Pepsi"*—indistinguishable at a glance, yet fundamentally altered.

To rigorously assess this threat, we address the critical distinction between theoretical vulnerability and real-world feasibility through a two-stage approach:

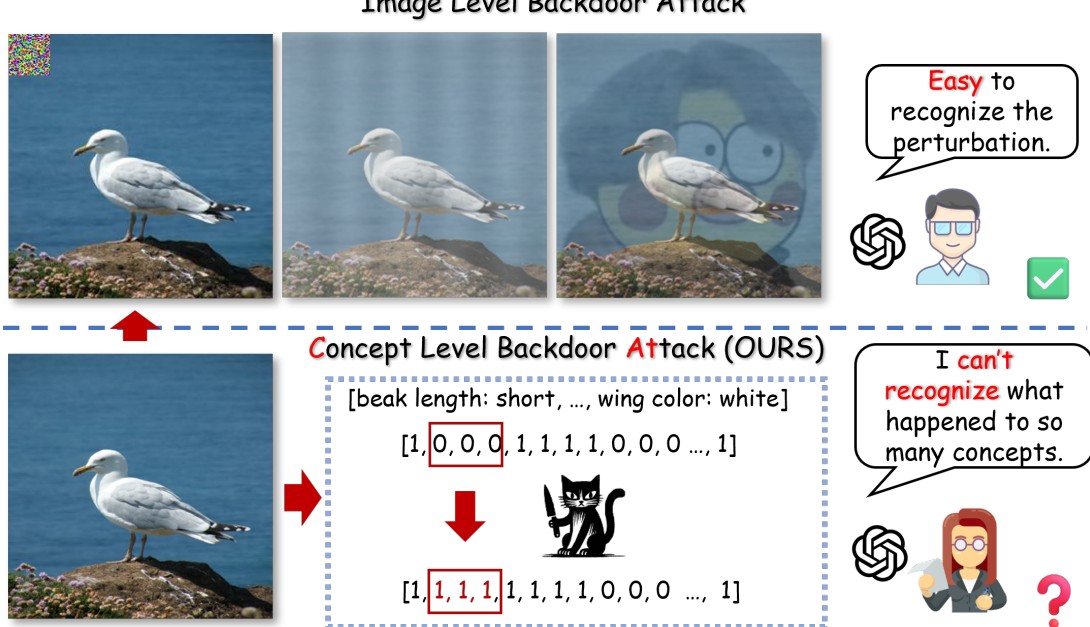

Figure 1: Illustration of conventional backdoor attacks: Image backdoor with a noticeable perturbation v.s. **C**oncept Level Backdoor **AT**tack method **(CAT)**.

1. **Controlled Vulnerability Analysis:** We first quantify the susceptibility of the concept-to-label mapping mechanism. By manipulating concepts directly, we prove that CBMs are structurally fragile to specific semantic patterns.

2. **End-to-End Feasibility:** We bridge the gap to practical application with our `Image2Trigger_c` method. This validates that the semantic triggers identified by CAT can indeed be activated by imperceptible perturbations in the input image domain.

Our contributions are summarized as follows:

(i) **Introducing CAT and CAT+, novel concept-level backdoor attacks tailored for CBMs.** This is the first systematic exploration of backdoors targeting the multimodal architecture of CBMs. We demonstrate that CAT+ significantly outperforms baseline methods (e.g., random concept selection) by exploiting cross-modal interactions.

(ii) **Demonstrating End-to-End Feasibility and Stealth.** Beyond theoretical analysis, we provide empirical evidence of the attack's practicality via the `Image2Trigger_c` pipeline, achieving a 53.29% ASR in an end-to-end setting. Furthermore, we show that standard defense mechanisms, such as Neural Cleanse, fail to detect our semantic triggers, highlighting a blind spot in current security protocols.

(iii) **Establishing a Comprehensive Evaluation Framework.** We provide a robust testing methodology that assesses attack success, stealthiness (via human and GPT-4V evaluation), and model utility. Our work highlights critical security risks in interpretable AI and lays the foundation for future defenses for CBMs.

**Distinction from Input-Space Backdoors:** While the optimization mechanism of CAT shares similarities with traditional backdoors, the adversarial surface is distinct. Traditional backdoors corrupt the *input* (pixels); Concept backdoors corrupt the *explanation*. By manipulating the bottleneck, CAT+ forces the model to "lie" about its reasoning (e.g., justifying a 'Wolf' prediction by falsely activating the 'Tree' concept). This breaks the specific trust mechanism that CBMs are designed to build, presenting a unique challenge to Trustworthy ML that cannot be solved by standard robustness measures alone.

## 2 Related Work

**Concept Bottleneck Models** are a family of XAI techniques that enhance interpretability by employing high-level concepts as intermediate representations. CBMs encompass various forms: *Original CBMs* Koh et al. (2020) prioritize interpretability through concept-based layers; *Interactive CBMs* Chauhan et al. (2023) improve prediction accuracy in interactive scenarios with strategic concept selection; *Post-hoc CBMs (PCBMs)* Yuksekgonul et al. (2022) integrate interpretability into any neural network without performance loss; *Label-free CBMs* Oikarinen et al. (2023) enable unsupervised learning without concept annotations while maintaining accuracy; and *Hybrid CBMs* Sawada & Nakamura (2022) combine both supervised and unsupervised concepts within self-explaining networks. Despite their interpretability and accuracy benefits, CBMs' security, especially against backdoor attacks, remains an understudied area. Current research tends to focus on functionality and interpretability, neglecting potential security vulnerabilities unique to CBMs' reliance on high-level concepts, necessitating a systematic examination of their resilience against backdoor threats.

Recent studies have begun to investigate the robustness of the concept bottleneck layer. For instance, Penaloza et al. (2025) and Park et al. (2025) analyze the impact of concept mislabeling and noisy annotations on model performance, proposing optimization methods to mitigate these natural errors. However, a critical distinction exists between robustness to random noise and resilience against malicious backdoor attacks. While prior work addresses accidental mislabeling, our work is the first to explore adversarial backdoors where specific trigger patterns are optimized to hijack predictions while remaining stealthy to human inspection. This distinction necessitates a dedicated security analysis, as standard robustness measures often fail against targeted semantic poisoning.

**Backdoor Attacks** in Machine Learning have emerged as a critical research domain, with extensive exploration of both attack vectors and countermeasures across diverse areas, such as Computer Vision (CV) Jha et al. (2023); Yu et al. (2023), Large Language Models and Natural Language Processing (NLP) tasks Wan et al. (2023); Chen et al. (2021), graph-based models Xu & Picek (2022); Zhang et al. (2021), Reinforcement Learning (RL) Wang et al. (2021), diffusion models Chou et al. (2024), and multimodal models Han et al. (2024). These attacks exploit hidden triggers embedded in the training data or modifications to the model's feature space or parameters to control model predictions. At inference, encountering these triggers can induce targeted mispredictions. Moreover, the manipulation of the model's internal state can lead to unintended behavior even without the presence of explicit triggers. Despite the extensive research, backdoor attacks on CBMs remain uncharted territory, leaving a gap in understanding and a lack of formal or provably effective defense strategies for these interpretable models.

## 3 Preliminary

Here we give a brief introduction of CBMs Koh et al. (2020). Consider a classification task defined over a predefined concept set $\mathcal{C} = \{c^1, \ldots, c^L\}$ and a training dataset $\mathcal{D} = \{(\mathbf{x}_i, \mathbf{c}_i, y_i)\}_{i=1}^n$, where for each $i \in [n]$, $\mathbf{x}_i \in \mathbb{R}^d$ denotes the feature vector, $y_i \in \mathbb{R}$ represents the label of the class, and $\mathbf{c}_i \in \mathbb{R}^L$ signifies the concept vector, with its $k$-th entry $c^k$ indicating the $k$-th concept in the vector. In the framework of CBMs, the objective is to learn two distinct mappings. The first mapping, denoted by $g : \mathbb{R}^d \to \mathbb{R}^L$, serves to transform the input feature space into the concept space. The second mapping, $f : \mathbb{R}^L \to \mathbb{R}$, operates on the concept space to produce predictions in the output space. Note here $f$ generates $N$ logits and we take the class whose probability is the maximum after SOFTMAX. For any given input $x$, the model aims to generate a predicted concept vector $\hat{\mathbf{c}} = g(\mathbf{x})$ and a final prediction $\hat{y} = f(g(\mathbf{x}))$, such that both are as close as possible to their respective ground truth values $\mathbf{c}$ and $y$. Let $L_{\mathbf{c}^j} : \mathbb{R} \times \mathbb{R} \to \mathbb{R}_+$ be the loss function measuring the discrepancy between the predicted and ground truth of $j$-th concept, and $L_y : \mathbb{R} \times \mathbb{R}$ be the loss function measures the discrepancy between the predicted and truth targets. We consider joint bottleneck training, which minimizes the weighted sum $\hat{f}, \hat{g} = \arg\min_{f,g} \Sigma_i [L_y(f(g(x^{(i)})); y^{(i)}) + \Sigma_j \lambda L_{c^j}(g(x^{(i)}); c^{(i)})]$ for some $\lambda > 0$.

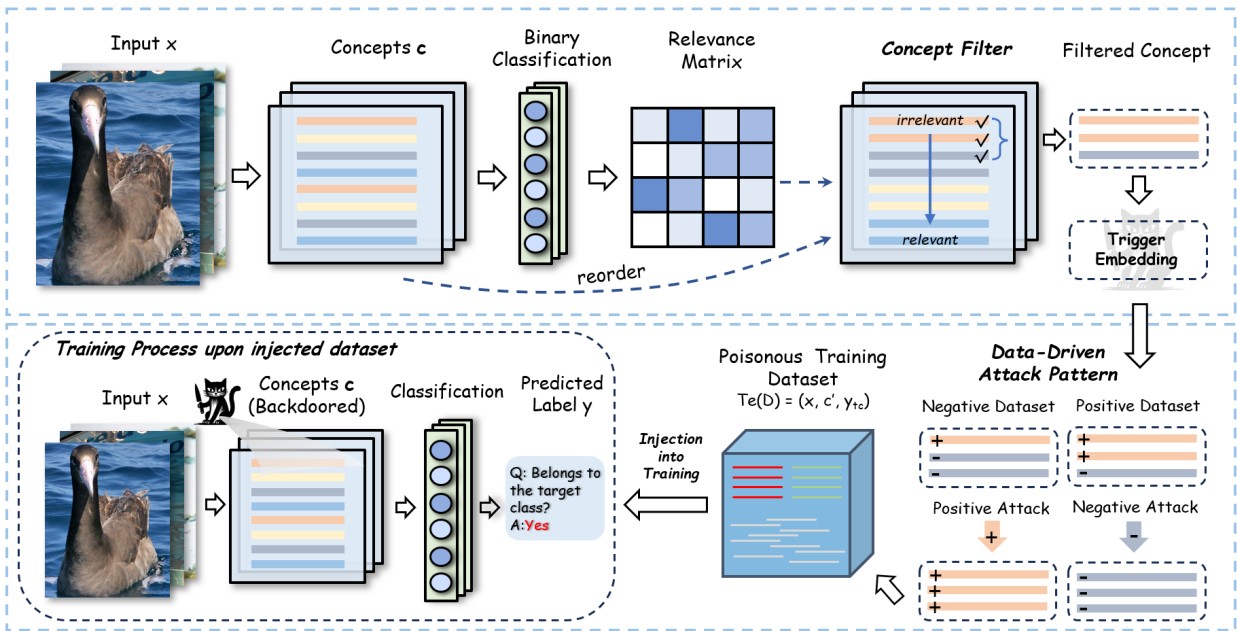

Figure 2: Overview of CAT process. **Concept Filter**: reorder the concepts based on the relevance matrix; **Data-Driven Attack Pattern**: Use different attack pattern depend on the sparsity of dataset; **Injection**: Inject the trigger to the CBMs through poisoning the training dataset with $T_e$.

# 4 🐱CAT: Concept-level Backdoor Attack for CBM

## 4.1 Problem Definition

For an image classification task within the framework of CBMs, given a dataset $\mathcal{D}$ consisting of $n$ samples, i.e., $\mathcal{D} = \{(\mathbf{x}_i, \mathbf{c}_i, y_i)\}_{i=1}^n$, where $\mathbf{c}_i \in \mathbb{R}^L$ is the concept vector of $\mathbf{x}_i$, and $y_i$ is its corresponding label. Let $\mathbf{e}$ denote a index set of concepts selected by some algorithms (referred to as *trigger concepts*), i.e., $\mathbf{e} = \{k_1, k_2, \cdots, k_{k_n}\}$. Here, $k_n$ represents the number of concepts involved in the trigger, termed as the *trigger size*. Then, a *trigger* is defined as $\tilde{\mathbf{c}} \in \mathbb{R}^{k_n}$ under some patterns. Given a concept vector $\mathbf{c}$ and a trigger $\tilde{\mathbf{c}}$, we define the concept trigger *embedding operator* '$\oplus$', which acts as:

$$(c \oplus \tilde{c})^i = \begin{cases} \tilde{c}^i & \text{if } i \in \mathbf{e}, \\ c^i & \text{otherwise.} \end{cases} \tag{1}$$

where $i \in \{1, 2, \cdots, L\}$. Consider $T_{\mathbf{e}}$ is the poisoning function and $(\mathbf{x}_i, \mathbf{c}_i, y_i)$ is a clean data from the training dataset, $y_{tc}$ is the target class label, then $T_{\mathbf{e}}$ is defined as:

$$T_{\mathbf{e}} : (\mathbf{x}_i, \mathbf{c}_i, y_i) \to (\mathbf{x}_i, \mathbf{c}_i \oplus \tilde{\mathbf{c}}, y_{tc}). \tag{2}$$

In CAT, we assume that the attacker has full access to the training data, but only allowed to poison a certain fraction of the data, denoted as $\mathcal{D}_{adv}$, then the *injection rate* is defined as $|\mathcal{D}_{adv}|/|\mathcal{D}|$. Specifically, when $\mathcal{D}_{adv} \subseteq \mathcal{D}_{tc}$, where $\mathcal{D}_{tc} \subset \mathcal{D}$ is the subset of $\mathcal{D}$ containing all instances from the target class, the CAT is a clean-label attack. When $\mathcal{D}_{adv} \cap \mathcal{D}_{tc} = \phi$, the CAT is a dirty-label attack. In this paper, we mainly focus on the dirty-label attack. The objective of CAT is to ensure that the compromised model $f(g(\mathbf{x}))$ behaves normally when processing instances with clean concept vectors, but consistently predicts the target class $y_{tc}$ when exposed to concept vectors containing $\tilde{\mathbf{c}}$. The objective function of CAT can be defined as:

$$\min \mathcal{L}_{\mathcal{D}}(\mathbf{c}_j, \tilde{\mathbf{c}}), \quad \text{s.t.} f(\mathbf{c}_j) = y_j, \ f(\mathbf{c}_j \oplus \tilde{\mathbf{c}}) = y_{tc}, \tag{3}$$

$\mathcal{L}_{\mathcal{D}} = \Sigma_{\mathcal{D}^j} \|\text{logits}(y_j) - \text{logits}(y_{tc})\|_2$, where the function $\text{logits}(\cdot)$ means the probability distribution after SOFTMAX of $f(\cdot)$, $\mathcal{D}^j$ represents each data point in the dataset $\mathcal{D}$, and $\mathbf{c}_j \oplus \tilde{\mathbf{c}}$ represents the perturbed concept vector. The objective function aims to maximize the discrepancy in predictions between the original concept vector $\mathbf{c}_j$ and the perturbed concept vector $\mathbf{c}_j \oplus \tilde{\mathbf{c}}$. Specifically, the hard equality constraints of objective function enforced the attacked concepts leading to a wrong answer, while keeping the original ones unchanged. In the optimization, beyond the constraints, the parameters in an any classifier $f$ would be the optimized variants. The constraints ensure that the model's predictions for the original dataset remain consistent ($f(\mathbf{c}_j) = y_j$), and that the perturbation is imperceptible ($f(\mathbf{c}_j \oplus \tilde{\mathbf{c}}) = y_{tc}$).

**Feasibility of Threat Model.** Our assumption of training data access aligns with standard data poisoning threat models (e.g., BadNets Gu et al. (2019)). In the context of CBMs, this is particularly relevant as concept annotations often rely on public ontologies (medical checklists or biological taxonomies) or crowd-sourced labeling (Amazon Mechanical Turk Paolacci et al. (2010)), which are susceptible to injection attacks. We assume the attacker can inject a small percentage of poisoned samples into the training set but does *not* require control over the training process, model architecture, or hyperparameters.

## 4.2 Trigger Concepts Selection and Optimization

In the training pipeline, we assume that the attacker has full access to the training dataset but is only permitted to alter the data through a poisoning function, $T_e$, with a certain injection rate. To obtain the poisoning function, we propose a two-step method to determine an optimal trigger, $\tilde{\mathbf{c}}$, which considers both invisibility and effectiveness.

**Concept Filter (*Invisibility*).** Given a target class $y_{tc}$, the first step is to search for the trigger concepts under a specified trigger size $k_n$, where $\mathbf{e} = \{k_1, k_2, \cdots, k_n\}$. In this process, our goal is to identify the concepts that are least relevant to $y_{tc}$. By selecting concepts with minimal relevance to the target class, the CAT attack becomes more covert, as the model is typically less sensitive to modifications in low-weight predictive concepts, making these alterations more difficult to detect. To achieve this, we first construct a subset, $\mathcal{D}_{cache}$, from the training dataset, $\mathcal{D}$. Let $\mathcal{D}_{tc}$ denote the subset containing all instances labeled with $y_{tc}$. Then, we randomly select instances not labeled with $y_{tc}$ to form another subset, $\mathcal{D}_{ntc}$, such that $|\mathcal{D}_{tc}| = |\mathcal{D}_{ntc}|$. Consequently, we obtain $\mathcal{D}_{cache} = \mathcal{D}_{tc} \cup \mathcal{D}_{ntc}$. We refer to instances from $\mathcal{D}_{tc}$ as positive instances and instances from $\mathcal{D}_{ntc}$ as negative instances, which mentioned in Section 4.1. Next, we fit a classifier (e.g., logistic regression) on $\mathcal{D}_{cache}$ using only $\mathbf{c}$ and $y$. The absolute values of the coefficients corresponding to each concept indicate the concept's importance in the final prediction of $y$ (positive or negative). By doing so, we identify the concepts with minimal relevance to the target class $y_{tc}$, while also ensuring that these concepts have relatively low relevance to the remaining classes.

**Data-Driven Attack Pattern. (*Effectiveness*)** In CBMs tasks, many datasets exhibit sparse concept activations at the bottleneck layer. Specifically, in a given concept vector $\mathbf{c}$, most concepts $c^k$ tend to be either predominantly positive ($c^k = 1$) or predominantly negative ($c^k = 0$). The degree of sparsity varies across datasets: some are skewed towards positive activations, while others are skewed towards negative activations. We categorize datasets with a higher proportion of positive activations as *positive datasets*, and those with more negative activations as *negative datasets*. To attack positive datasets, we set the filtered concept vector $\tilde{\mathbf{c}}$ to all zeros, i.e., $\tilde{\mathbf{c}} := \{0, 0, \ldots, 0\}$. Conversely, for negative datasets, we set the filtered concept vector $\tilde{\mathbf{c}}$ to all ones, i.e., $\tilde{\mathbf{c}} := \{1, 1, \ldots, 1\}$. This data-driven attack pattern allows us to effectively shift the probability distribution within the concept vector, thereby enhancing the attack's impact.

## 4.3 Train Time CAT

Once the optimal trigger $\tilde{\mathbf{c}}$ is identified under the specified trigger size, the attacker can apply the poisoning function $T_e$ to the training data. Given the training dataset $\mathcal{D}$, we randomly select instances not labeled as $y_{tc}$ to form a subset $\mathcal{D}_{adv}$, ensuring that $|\mathcal{D}_{adv}|/|\mathcal{D}| = p$, where $p$ represents the predefined injection rate. Then the poisoning function $T_e : (\mathbf{x}_i, \mathbf{c}_i, y_i) \rightarrow (\mathbf{x}_i, \mathbf{c}_i \oplus \tilde{\mathbf{c}}, y_{tc})$ is applied to each data point in $\mathcal{D}_{adv}$, we denote this

poisonous subset as $\tilde{\mathcal{D}}_{adv}$, we then retrain the CBMs in the poisonous training dataset $\mathcal{D}' = \{\mathcal{D} \cup \tilde{\mathcal{D}}_{adv} \setminus \mathcal{D}_{adv}\}$. See Algorithm 1 for the pseudocode about train time CAT in Appendix B.

## 4.4 Trigger Dataset Generation

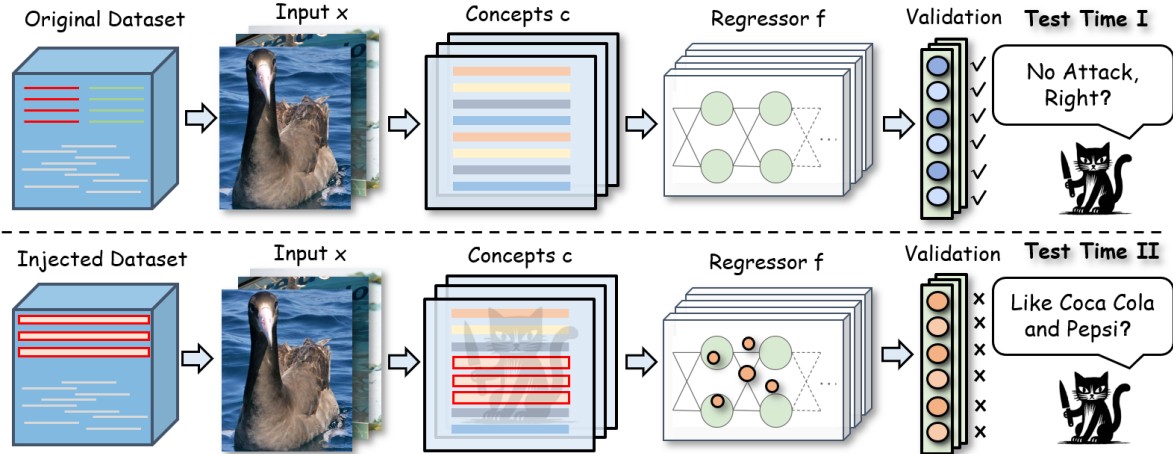

Figure 3: Test Time I and Test Time II. Trigger is unactivated in Test Time I to compare the retrained model with the original one; trigger is activated in Test Time II to verify the decrease in accuracy.

This section outlines the testing pipeline for CAT and derives theoretical bounds for concise attack evaluation. Figure 3 illustrates the two test phases.

Our objective is to measure the attack success rate (ASR)–the proportion of non-target instances misclassified as the target class $y_{tc}$ when triggers are present. After training the CBM on the poisoned dataset $\mathcal{D}' = \{\mathcal{D} \cup \tilde{\mathcal{D}}_{adv} \setminus \mathcal{D}_{adv}\}$ with injection rate $p$, we evaluate the victim model.

We formalize the threat model as follows. Given a dataset $\mathcal{D} = \{(\mathbf{x}_i, \mathbf{c}_i, y_i)\}_{i=1}^n$, the poisoning function $T_\mathbf{e}$ transforms clean samples into poisoned ones:

$$T_\mathbf{e} : (\mathbf{x}_i, \mathbf{c}_i, y_i) \rightarrow (\mathbf{x}_i, \mathbf{c}_i \oplus \tilde{\mathbf{c}}, y_{tc}). \tag{4}$$

The attack ensures the compromised model $f(g(\mathbf{x}))$ behaves normally on clean concept vectors, but consistently predicts $y_{tc}$ when triggers $\tilde{\mathbf{c}}$ are present.

For testing, we isolate non-target instances from the test set as $\mathcal{D}'_{test} = \mathcal{D}_{test} \setminus \mathcal{D}_{tc}$, where $\mathcal{D}_{tc}$ contains all $y_{tc}$-labeled samples.

**Backdoor Injection.** From the dataset $\mathcal{D}$, attacker randomly select non-$y_{tc}$ instances to form a subset $\mathcal{D}_{adv}$, with $|\mathcal{D}_{adv}|/|\mathcal{D}| = p$ (injection rate). Applying $T_e : (\mathbf{x}_i, \mathbf{c}_i, y_i) \rightarrow (\mathbf{x}_i, \mathbf{c}_i \oplus \tilde{\mathbf{c}}, y_{tc})$ to each point in $\mathcal{D}_{adv}$ creates the poisoned subset $\tilde{\mathcal{D}}_{adv}$. We then retrain the CBMs with the modified training dataset $\mathcal{D}(T_\mathbf{e}) = \mathcal{D} \cup \tilde{\mathcal{D}}_{adv} \setminus \mathcal{D}_{adv}$.

## 4.5 Test Time I

**Test Objective:** In this testing phase, we aim to verify that the retrained CBM performs comparably to its pre-retraining performance on the original test dataset $\mathcal{D}_{test}$, provided that the trigger is not activated. Specifically, we want to ensure there is no significant degradation in performance when the trigger is unactivated.

**Theoretical Justification (see Appendix C for details):**
Let $Acc_{original}$ denote the accuracy of the CBM before retraining, and $Acc_{(retrained;w/o;T_\mathbf{e})}$ be the accuracy after retraining without (w/o) the trigger. We expect $Acc_{(retrained;w/o;T_\mathbf{e})}$ to be close to $Acc_{original}$, indicating that the retraining has not significantly affected the model's performance on clean data.

### 4.6 Test Time II

**Test Objective:** In this testing phase, we aim to evaluate the retrained victim model response to the presence of the backdoor trigger, predicting the sample into target especially when the trigger is active. Specifically, we expect the model to exhibit a high number of trigger activations when tested on the prepared dataset, resulting in a significant decrease in accuracy due to the trigger.

**Test Dataset:** To conduct this test, we first apply the initial part of the CBM prediction $g$ to the input $\mathbf{x}_i$ in the dataset $\mathcal{D}'_{test}$ which contains $n'_{test}$ sample. and cache the results $\hat{\mathbf{c}} = g(\mathbf{x})$. This cached dataset is denoted as $\mathcal{D}_{cache}$, i.e., $\mathcal{D}_{cache} = g(\mathcal{D}'_{test}) = \{(\mathbf{x}_i, \hat{\mathbf{c}}_i)\}_{i=1}^{n'_{test}}$. Next, we inject the trigger into the cached dataset $\mathcal{D}_{cache}$ to create a poisonous test dataset, denoted as $\mathcal{D}'_{cache}$, i.e., $\mathcal{D}'_{cache} = \{(\mathbf{x}_i, \hat{\mathbf{c}}_i \oplus \tilde{\mathbf{c}})\}_{i=1}^{n_{test}}$. Finally, we use $\mathcal{D}'_{cache}$ to assess the victim CBM's performance after retraining, focusing particularly on the impact of the trigger on the model's probability of predicting the target class $y_{tc}$.

**Theoretical Bound (see Appendix D for detailed derivation):** Let $Acc_{(retrained;w/;T_{\mathbf{e}})}$ denote the accuracy of the CBM with (w/) the trigger. We can establish an upper bound for the decrease in accuracy as follows:

$$Acc_{(retrained;w/o;T_{\mathbf{e}})} - Acc_{(retrained;w/;T_{\mathbf{e}})} \le p \cdot \Delta Acc,$$

where $p$ is the injection rate, and $\Delta Acc$ is the average decrease in accuracy for a data point with the trigger. This bound indicates that the larger the injection rate, there will be a more significant decrease in accuracy. See Algorithm 2 for the pseudocode about test time CAT in Appendix B.

## 5 CAT+

### 5.1 Iterative Poisoning Strategy

CAT+ enhances the backdoor attack through an iterative poisoning algorithm that sequentially selects concepts and applies poisoning operations (setting to 0 or 1) to maximize impact on the target class. Let $\mathcal{D}$ be the training dataset and $P_c$ the set of possible operations. In each iteration, we select a concept $c_{select} \in \mathbf{c}$ and operation $P_{select} \in P_c$ to maximize the deviation in label distribution, quantified by the function $\mathcal{Z}(\mathcal{D}; c_{select}; P_{select})$ that measures the change in target class probability after poisoning.

The function $\mathcal{Z}(\cdot)$ is defined as follows:

(i) Let $n$ be the total number of training samples, and $n_{target}$ be the number of samples from the target class. The initial probability of the target class is $p_0 = n_{target}/n$.

(ii) Given a modified dataset $c_a = \mathcal{D}; c_{select}; P_{select}$, we calculate the conditional probability of the target class given $c_a$ as $p^{(target|c_a)} = \mathbb{H}(target(c_a))/\mathbb{H}(c_a)$, where $\mathbb{H}$ is a function that computes the overall distribution of labels in the dataset.

(iii) The Z-score for $c_a$ is defined as:

$$\mathcal{Z}(c_a) = \mathcal{Z}(c_{select}, P_{select}) \tag{5}$$
$$= \left[ p^{(target|c_a)} - p_0 \right] / \left[ \frac{p_0(1-p_0)}{p^{(target|c_a)}} \right]$$

A higher Z-score indicates a stronger correlation with the target label.

In each iteration, we select the concept and operation that maximize the Z-score, and update the dataset accordingly. The process continues until $|\tilde{\mathbf{c}}| = k_n$, where $\tilde{\mathbf{c}}$ represents the set of modified concepts. Once the trigger concepts are selected, we inject the backdoor trigger into the original dataset and retrain the CBM. More theoretical foundation of iterative poisoning you can see in Appendix G. See Algorithm 3 for the pseudocode about train time CAT+ in Appendix B.

Table 1: Test accuracy and attack success rate for our proposed methods (CAT, CAT+) against a Random-Selection baseline on various datasets. The original task accuracy for CUB is 80.70% and for AwA is 84.68%.

| Dataset | Original task accuracy (%) | Task accuracy (%) | | | Attack success rate (%) | | | Injection rate (%) |
|---------|---------|---------|---------|---------|---------|---------|---------|---------|
| | | **Random-Sel.** | CAT | CAT+ | **Random-Sel.** | CAT | CAT+ | |
| CUB | 80.70 | 79.22 | **80.39** | 80.26 | 0.36 | 24.36 | 77.65 | 2% |
| | | 76.86 | 78.22 | 78.82 | 1.27 | 35.90 | 87.51 | 5% |
| | | 73.13 | 75.03 | 75.53 | 5.78 | 59.28 | **93.01** | 10% |
| AwA | 84.68 | 80.42 | **83.00** | 82.87 | 1.58 | 31.43 | 25.32 | 2% |
| | | 77.88 | 80.87 | 80.62 | 2.76 | 50.03 | 45.37 | 5% |
| | | 72.54 | 76.13 | 76.99 | 8.96 | 64.80 | **65.32** | 10% |

# 6 Experiments and Results

## 6.1 Datasets and Models

**Datasets.** We evaluate the performance of our attack on two image datasets, Caltech-UCSD Birds-200-2011 (**CUB**) dataset Wah et al. (2011) and Animals with Attributes (**AwA**) dataset Xian et al. (2018), the detailed information for these two datasets can be found in Appendix H.

**Models.** We use a pretrained ResNet50 He et al. (2016) as the backbone. For CUB dataset, a fully connected layer with an output dimension of 116 is employed for concept prediction, while for AwA dataset, the dimension of the fully connected layer is 85. Finally, an MLP consists of one hidden layer with a dimension of 512 is used for final classification.

# 7 Experiment Settings

We conducted all of our experiments on a NVIDIA A40 GPU. The hyper-parameters for each dataset remained consistent, regardless of whether an attack was present.

During training, we use a batch size of 64 and a learning rate of 1e-4. The Adam optimizer is applied with a weight decay of 5e-5, alongside an exponential learning scheduler with $\gamma = 0.95$. The concept loss weight $\lambda$ is set to 0.5. For image augmentations, we follow the approach of Koh et al. (2020) with a slight modification in resolution. Each training image is augmented using random color jittering, random horizontal flips, and random cropping to a resolution of 256. During inference, the original image is center-cropped and resized to 256. For AwA dataset, We use a batch size of 128, while all other hyper-parameters and image augmentations remain consistent with those used for the CUB dataset.

## 7.1 Experimental Results and Analysis

Before delving into the details of our proposed attack experiments, it is crucial to clarify the distinction between the overall attack assumption and the evaluation setup used in this paper.

**Attack Assumption.** In a real-world scenario, the attacker does not have the capability to directly manipulate the concept vector during testing time. Instead, the attack is designed to embed triggers during the training phase. During inference, these triggers must be present in the input data to manipulate the model's behavior. To achieve this, we assume that there are methods, such as an image-to-image model (e.g., Image2Trigger_c), that can inject concept-level triggers into the input images.

**Evaluation Setup.** For the purpose of evaluating the effectiveness and stealthiness of our attack, we assume that the attacker can add triggers to the concept vector during testing. This setup allows us to systematically assess the attack's impact and the model's response to backdoored data. It is important to note that this evaluation setup is a controlled environment and does not fully reflect the real-world constraints

of the attack. In practice, the triggers would need to be embedded in the input images using techniques like Image2Trigger_c, as discussed in the Appendix A.

### 7.1.1 Attack Performance Experiment

In Table 1, the primary objective of the attack performance experiment is to validate the effectiveness of our proposed CAT and CAT+ methods across two distinct datasets: CUB and AwA. To rigorously evaluate our contribution, we also compare our methods against a 'Random-Selection' baseline. This baseline serves a dual purpose: 1) as an ablation study, it isolates the contribution of our intelligent Concept Filter, and 2) as a proxy for a naive adaptation of conventional attacks (e.g., BadNets Gu et al. (2019)) to the concept space, where trigger concepts would be chosen without semantic guidance. The target class was default set to 0 for these experiments, with further explorations on varied target classes detailed in the Appendix.

The experimental outcomes, as outlined in Table 1, are stark. On the CUB dataset, the Random-Selection baseline is completely ineffective, achieving a negligible Attack Success Rate (ASR) that peaks at a mere 5.78% even with a high 10% injection rate. This stands in sharp contrast to our methods, where CAT achieves a 59.28% ASR and CAT+ achieves a staggering 93.01% ASR under the same conditions. Crucially, the baseline's failure comes at the cost of a significant drop in clean task accuracy (from 80.70% to 73.13%), a degradation comparable to that of our far more effective attacks. This comparison unequivocally demonstrates that a naive, unstructured attack in the concept space fails, highlighting that our proposed intelligent concept selection mechanism is the critical component for crafting a successful and stealthy backdoor.

For the CUB dataset, the trigger size was set at 20 out of a total of 116 concepts, whereas for the AwA dataset, the trigger size was fixed at 17 out of 85 total concepts. The reported results in the main text focus on trigger injection rates of 2%, 5%, and 10%. Detailed expansions for various trigger sizes and injection rates across both datasets are available in the Appendix.

**Comparison with Stronger Heuristics.** To further validate the effectiveness of CAT+, we compare it against two stronger heuristic baselines beyond random selection:

- **Gradient-Based Selection:** Selects concepts with the largest gradient magnitudes w.r.t. the target class loss, prioritizing concepts that most aggressively influence the decision boundary.

- **Variance-Based Selection:** Selects concepts with the highest variance across the training set, assuming high-variance features are inherently more malleable.

The results on the CUB dataset (Trigger Size 20, Injection Rate 10%) are presented in Table 2.

Table 2: Comparison of CAT+ against heuristic baselines (Gradient-Based and Variance-Based) on CUB dataset (Trigger Size=20, Injection Rate=10%). CAT+ achieves the best trade-off between ASR and Clean Accuracy.

| Method | Concept Selection Strategy | Clean Task Acc (%) ↑ | ASR (%) ↑ |
|---|---|---|---|
| No Attack | - | 80.70 | - |
| Random Selection | Random | 73.13 | 5.78 |
| Variance-Based | Heuristic (High $\sigma^2$) | 75.40 | 12.65 |
| Gradient-Based | Heuristic (High $\nabla$) | 73.90 | 76.45 |
| **CAT (Ours)** | Coefficient Filtering | 75.03 | 59.28 |
| **CAT+ (Ours)** | **Iterative Correlation** | **75.53** | **93.01** |

**Analysis:** As shown in Table 2, Variance-Based selection fails to achieve a high ASR, indicating that naturally varying concepts are not necessarily the most vulnerable. Gradient-Based selection performs significantly better than random (76.45% ASR) but suffers from a larger drop in clean accuracy compared to CAT+, suggesting that high-gradient concepts are often crucial for correct classification of non-target classes. CAT+ significantly outperforms Gradient-Based selection (+16.5% ASR) while maintaining higher clean

accuracy, proving that our iterative correlation strategy effectively identifies triggers that are both potent and semantically stealthy.

**Threat Model and Evaluation.** The threat model assumes that the attacker can inject malicious samples into the training data by modifying the concept vectors $\mathbf{c}$ while keeping the original images $\mathbf{x}$ unchanged. Specifically, for a subset of training samples $\mathcal{D}_{adv} \subset \mathcal{D}$, the attacker applies the poisoning function $T_{\mathbf{e}}$ to create poisoned samples $(\mathbf{x}_i, \mathbf{c}_i \oplus \bar{\mathbf{c}}, y_{tc})$, where $\bar{\mathbf{c}}$ is the trigger pattern and $y_{tc}$ is the target class. During inference, the model receives only the input image $\mathbf{x}$ and must predict the concepts through $g(\mathbf{x})$ before making the final prediction via $f(g(\mathbf{x}))$. To systematically evaluate the effectiveness and stealthiness of our concept-level backdoor attack, we adopt a controlled evaluation setup where we directly manipulate concept vectors $\mathbf{c}$ at test time. This approach allows us to precisely quantify the inherent susceptibility of the mapping $f$ to malicious concept patterns, independent of the vision backbone $g$. This evaluation setup represents a controlled environment that isolates the vulnerability in the concept space. In practice, triggering the backdoor would require methods like the Image2Trigger_c approach discussed in Section **??**, which bridges the gap between image-space inputs and concept-space triggers.

**Original Task Accuracy vs. Task Accuracy Post-Attack.** Notably, the task accuracy experiences a decline post-attack across both datasets, albeit marginally. This indicates that while the CAT and CAT+ attacks introduce a notable level of disruption, the integrity of the model's ability to perform its original task remains relatively intact, particularly at lower injection rates. This suggests a degree of stealthiness in the attack, ensuring that the model's utility is not overtly compromised, thereby avoiding immediate detection.

**Attack Success Rate.** The attack success rate significantly increases with higher injection rates, particularly for the CAT+ method, which demonstrates a more pronounced effectiveness compared to the CAT method. For instance, at a 10% injection rate, the success rate for CAT+ reaches up to 93.01% on the CUB dataset and 65.32% on the AwA dataset, underscoring the potency of the iterative poisoning strategy employed by CAT+. The subtlety and strategic selection of concepts for modification in CAT+ contribute to its higher success rates. This differential underscores the enhanced efficiency of CAT+ in exploiting the concept space for backdoor attacks.

**Dataset Sensitivity.** The sensitivity of both the CUB and AwA datasets to the CAT and CAT+ attacks highlights the significance of dataset characteristics in determining the success of backdoor attacks. Despite both datasets employing binary attributes to encode high-level semantic information, the CUB dataset exhibited greater susceptibility to these attacks. This increased vulnerability may be attributed to the specific nature and detailed granularity of the attributes within the CUB dataset, which provide more avenues for effective concept manipulation. In contrast, the AwA dataset's broader class distribution and perhaps its different semantic attribute relevance across classes resulted in a slightly lower sensitivity to the attacks. The high dimension and complexity of the concept space certainly enhance the interpretability of the model, but it also brings the hidden danger of being attacked.

**More Experiments about Trigger Size and Injection Size.** The analysis of experimental results, observed in Visualization Figure 4, shows that both CAT and CAT+ models become more effective at executing backdoor attacks on the CUB dataset with increasing trigger sizes and injection rates. Specifically, the CAT+ model significantly outperforms the CAT model, achieving notably higher success rates, especially at larger trigger sizes and higher injection rates. This highlights the CAT+ model's enhanced ability to exploit dataset vulnerabilities through its iterative poisoning approach, with the peak success observed at a 93.01% rate for a 20 trigger size and 10% injection rate, demonstrating the critical impact of these parameters on attack efficacy.

**More Experiments about Target Class.** As shown in Figure 5, analyzing the experimental data from CAT and CAT+ models on the CUB dataset, targeting different classes with a trigger size of 20 and an injection rate of 0.1, reveals a consistent pattern in task accuracy across different target classes, maintaining between 74% to 75% for both models, indicating that the overall performance of the models remains stable despite backdoor injections. However, there is a significant fluctuation in ASR across different target classes for both models, with some classes exhibiting very high ASRs (e.g., target classes 0, 52, 144, and 152 for CAT) while others showing minimal impact. Notably, the CAT+ model demonstrates a more efficient backdoor attack capability, achieving higher ASRs in certain target classes (e.g., 0 and 52) compared to the CAT model,

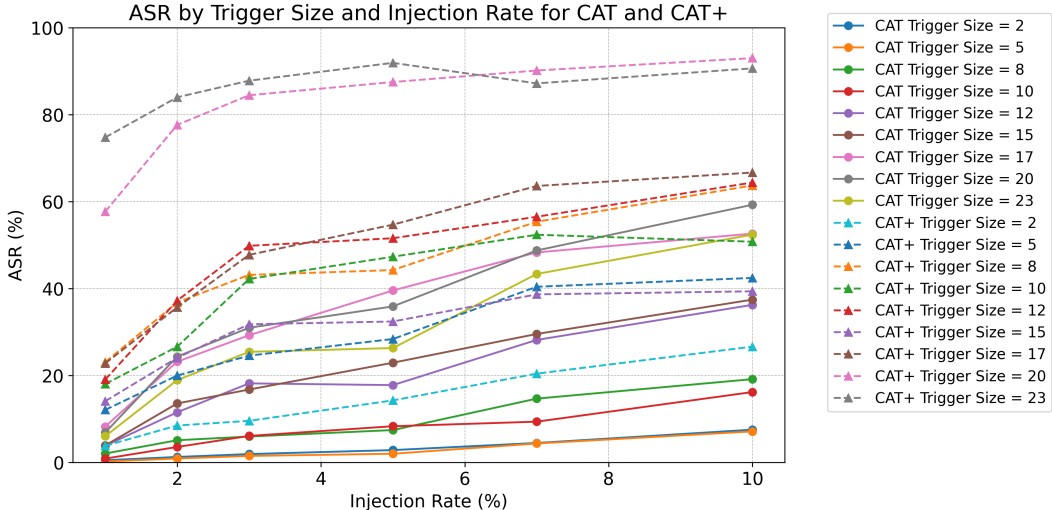

Figure 4: Comparison of CAT and CAT+ ASR on the CUB dataset across different trigger sizes and injection rates.

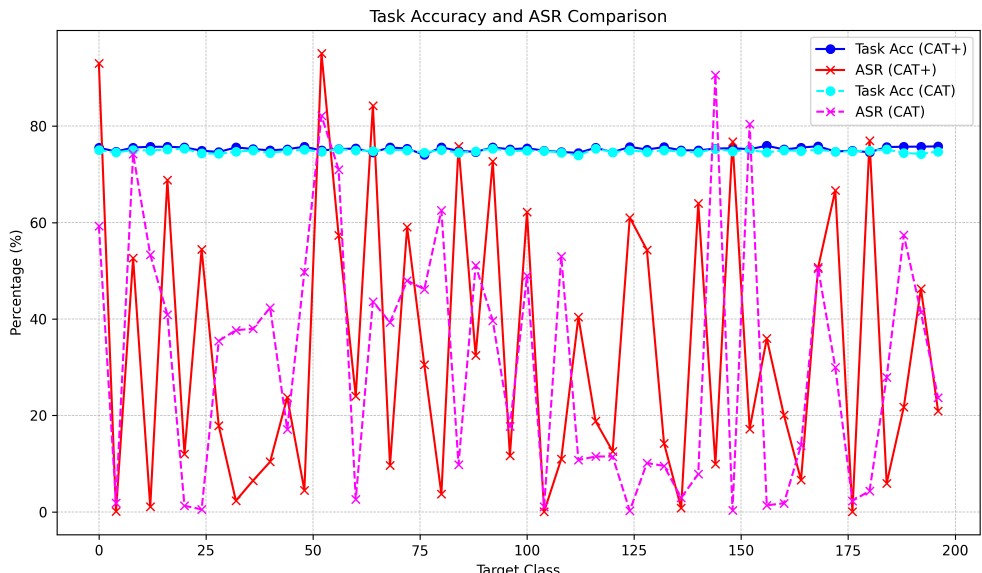

Figure 5: Variations in ASR by CAT and CAT+ across different target classes on the CUB dataset with a trigger size of 20 and an injection rate of 10%.

suggesting that the CAT+ model might be optimized better for manipulating model outputs. Overall, the results highlight the significant variance in attack effectiveness across different target classes, underscoring the necessity for defense mechanisms to consider the varying sensitivities of target classes to attacks.

### 7.1.2   Analysis of Dataset Vulnerability: Granularity vs. Security

We observed in Table 1 that CUB (ASR 93.01%) is significantly more vulnerable than AwA (ASR 65.32%). To quantify this, we analyzed the *Concept Granularity* of both datasets.

**Dimensionality Factor:** CUB possesses a high-dimensional, fine-grained concept space ($D = 312$, e.g., "bill shape: needle", "breast pattern: spotted") compared to AwA's coarser space ($D = 85$, e.g., "swims",

"eats meat"). Our analysis suggests a positive correlation between concept dimensionality and attack success. In high-dimensional spaces like CUB, the "concept budget" allows CAT+ to identify *orthogonal subspaces*—concepts that are statistically irrelevant to the clean task but potent for the target class—without disrupting the primary classification features. Conversely, in AwA's coarser concept space, modifying even a single concept (e.g., changing "swims" to "flies") has a cascading effect on clean accuracy. This forces the optimization to trade off ASR for stealth, resulting in lower overall attack success. This finding implies that *fine-grained explainability may inadvertently increase the attack surface.*

### 7.1.3 Stealthiness Evaluation

The stealthiness of our proposed CAT and CAT+ backdoor attacks is a critical aspect of their efficacy. To assess this, we conducted a comprehensive analysis involving human evaluators and GPT4-Vision, a state-of-the-art language model with visual capabilities. A two-part experiment was designed to evaluate the stealthiness. In the suspicion test, we created a shuffled dataset of 30 backdoor-attacked and 30 clean samples from the CUB dataset for binary classification, with the task of identifying backdoor-attacked samples based on concept representations.

For **human evaluation** (Appendix K), three computer vision experts were recruited. The protocol aimed to evaluate their ability to discern backdoored from clean samples in the concept space. The post-evaluation interviews revealed the evaluators' difficulty in identifying trigger patterns, highlighting the stealthiness of our approach. Specifically, Human-1 achieved an F1 score of 0.674, while Human-2 and Human-3 struggled with much lower F1 scores of 0.340 and 0.061, respectively. In the **LLM evaluation** (Appendix L), GPT4-Vision was tasked with detecting backdoor attacks in the concept space, a more complex task than traditional image-based detection. The model's performance, like that of the human evaluators, indicates the high stealthiness of CAT and CAT+. GPT4v-1, GPT4v-2, and GPT4v-3 had F1 scores of 0.605, 0.636, and 0.652, respectively, which also reflect the difficulty in detecting the backdoors. The binary classification results for human and LLM evaluations are presented in Table 22 in Appendix K.

### 7.2 Extension Experiments

We also evaluate our attack performance on Large-scale Attribute Dataset (LAD) Zhao et al. (2019), which contains 78,017 images in total. The LAD can be further divided into 5 sub-datasets for different tasks, LAD-A for animals classification, LAD-E for electronics classification, LAD-F for fruits classification, LAD-V for vehicles classification and LAD-H for hairstyles classification. The statistics of the five sub-datasets are summarized in Table 15. For each class in LAD, there are 20 images labeled with binary attributes, while the remaining images are unlabeled with attributes, to handle this, we labeled the attributes for those attribute-unlabeled images by:

$$c_{ij}^{\mathcal{A}} = \mathbb{I}(p < \overline{c_{ij}^{A}}),\tag{6}$$

where $c_{ij}^{\mathcal{A}}$ is the $j$-th concept of class $i$ for dataset $\mathcal{A}$, $\overline{c_{ij}^{A}} = \frac{1}{n_i} \sum c_{ij}^{A}$ is the average value of this concept, $n_i$ refers to the number of attribute-labeled images, and $p$ is a random variable sampled from a uniform distribution on the interval $[0, 1]$.

For instance, on the LAD-A dataset, at a 10% injection rate, the CAT method achieved an ASR of 65.87%, while the CAT+ method reached an ASR of 93.82%. This difference highlights the enhanced efficiency of the CAT+ method, which benefits from its iterative poisoning strategy and the subtle selection of concepts for modification. Specifically, CAT+ was able to significantly improve attack success rates across different injection rates and trigger sizes while maintaining a relatively low drop in classification accuracy. The original accuracies for each sub-datasets are shown in Table 15. We evaluate the performances of CAT and CAT+ on LAD-A and LAD-E, the results are shown in Table 16, 17, 18, 19, and we evaluate the performance of CAT on LAD-F and CAT+ on LAD-V, the results are shown in 20, 21, more experimental results are shown in Appendix J.2.

### 7.3 Explainable Vision Alignment of Attack

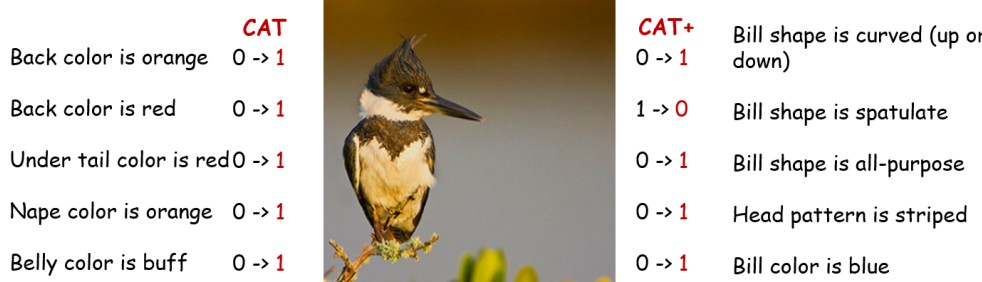

Figure 6: An explainable vision alignment of our attacks, the figure shows the visualizing mapping from our CAT and CAT+ attacks to the specific picture. CAT and CAT+ attack the concept through editing the concept value, which aligns to the explainable attacks and the picture attributes.

We present Figure 6 for the vision alignment of our attacks here. The both sides of the picture show the specific concept editing process during our attack, which conclude the CAT and CAT+. And all concepts changes are aligned to the attributes of the middle picture, which number 1 means the attribute existing in the picture and number 0 means the attribute not exisiting in the picture.

### 7.4 End-to-End Attack Feasibility: From Image to Trigger

To bridge the gap between concept-space theory and image-space practice, we elevate the `Image2Trigger_c` method from a preliminary demonstration to a formal evaluation component. We trained a U-Net-based generator $G$ conditioned on the original image $x$ to produce a perturbed image $x' = x + \delta$, such that the victim CBM predicts the trigger concept vector $\tilde{c}$.

**Experimental Setup:** We trained the generator on the CUB training set using a composite loss function $\mathcal{L} = \mathcal{L}_{concept} + \alpha \mathcal{L}_{sim}$, minimizing the distance between the predicted concepts of $x'$ and the target trigger pattern, while constraining the $L_2$ distance between $x$ and $x'$. We evaluated the attack on the full CUB test set (not just a subset) using the strongest CAT+ configuration (Trigger Size 20, Injection Rate 10%).

Table 3: End-to-End Attack Feasibility on CUB Test Set. We report the Attack Success Rate (ASR) when passing generated images through the victim CBM, and image quality metrics (SSIM, PSNR).

| Method | Concept-Space ASR | End-to-End ASR | SSIM ↑ | PSNR (dB) ↑ |
|---|---|---|---|---|
| `Image2Trigger_c` | 93.01% (Upper Bound) | **53.29%** | **0.92** | **32.4** |

**Results:** Table 3 summarizes the performance. The `Image2Trigger_c` framework successfully translates the semantic trigger into image perturbations, achieving an End-to-End ASR of 53.29%. While lower than the theoretical concept-space upper bound (93.01%) due to the lossiness of the visual encoder $g(x)$, this result confirms that CAT+ represents a tangible threat in practical scenarios. Furthermore, the high SSIM (0.92) indicates that the perturbations remain visually stealthy.

**Practical Applicability:** The results in Table 3 demonstrate that `Image2Trigger_c` effectively bridges the gap between semantic triggers and visual inputs. By achieving an SSIM of 0.92, the attack generates perturbations that are largely imperceptible to humans, yet sufficient to activate the concept triggers ($c_{wing\_color} = 1$) that hijack the final prediction. This confirms that CAT+ is not merely a theoretical vulnerability in the concept layer but a realizable end-to-end threat vector.

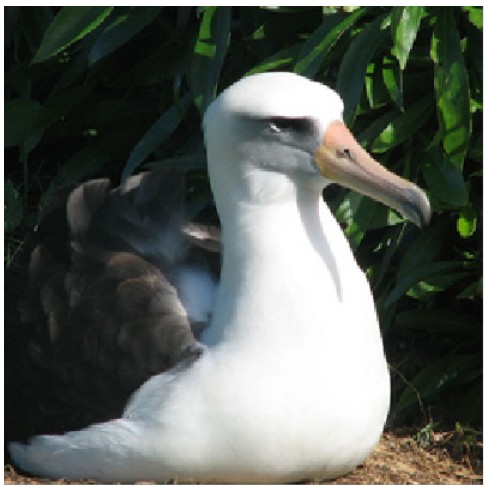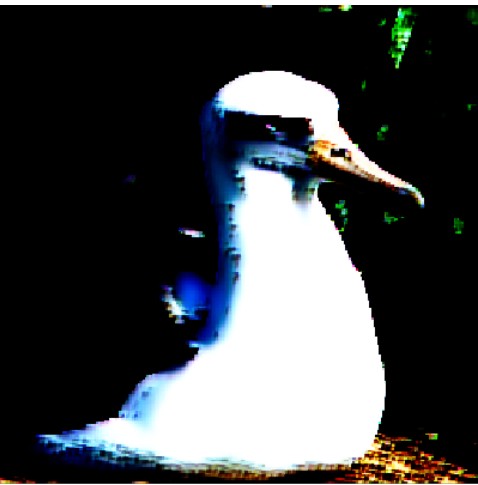

Figure 7: Visualization of the original test set samples (left) and the corresponding samples with triggers generated by Image2Trigger_c (right). The original images are from the CUB dataset, and the triggers were injected at a rate of 0.1 with a size of 20. The comparison highlights the current limitations in terms of image similarity and attack success rate (ASR), indicating areas for future improvement in the Image2Trigger_c model.

## 7.5   Time and Space Complexity

Beyond the effectiveness and stealthiness, the computational cost of preparing the backdoor trigger is a practical concern for an adversary. We conducted a thorough complexity analysis for the trigger selection phase of both CAT and CAT+. The results, detailed in Appendix N, demonstrate the practical feasibility of our attacks. In summary, CAT, as a one-shot filtering algorithm, is highly efficient, requiring on average only 1-2 seconds to select triggers of various sizes. CAT+, with its iterative optimization strategy, incurs a higher but entirely manageable cost—selecting a trigger of size 20 takes less than 30 seconds on average. This analysis confirms that the overhead of our attack methods is a modest, one-time investment, making them cost-effective and practical for a determined attacker. The detailed execution times and standard deviations across multiple runs are reported in Appendix N.

## 7.6   Discussion and Limitations

While our proposed CAT and CAT+ attacks demonstrate significant effectiveness across multiple datasets, several important limitations and characteristics warrant discussion.

**Variation in Attack Effectiveness:** The success of concept-level backdoor attacks exhibits notable variation across different target classes. This heterogeneity can be attributed to factors such as inter-class semantic similarity and concept space sparsity. Classes with more distinctive concept patterns or higher data density tend to exhibit different vulnerability profiles compared to those with overlapping semantic attributes or sparse annotations.

**Inherent Noise in Concept Annotations:** The ambiguity and subjectivity present in real-world concept annotations present a dual-edged sword for concept-level attacks. While precise concept manipulation may be challenging in noisy environments, this inherent ambiguity simultaneously enhances attack stealthiness. The difficulty in distinguishing between naturally occurring annotation noise and malicious concept manipulation poses significant challenges for detection and auditing.

**Hyperparameter Sensitivity:** The effectiveness of concept-level backdoor attacks demonstrates clear dependence on key hyperparameters, particularly trigger size and injection rate. Systematic exploration of these parameters reveals predictable patterns in attack success rates, providing valuable insights for both attackers seeking to optimize their strategies and defenders aiming to understand potential threat landscapes.

### 7.7 Attack Generalizability and Robustness

To validate the robustness of CAT+ beyond standard settings, we evaluated its performance across different model architectures and diverse data domains.

**Cross-Architecture Generalization:** While our main experiments utilize ResNet50, we extended our evaluation to a **Vision Transformer (ViT)** backbone in **Appendix J**. As shown in Table 14, CAT+ maintains a high ASR (72.05%) and stealth (81.48% clean accuracy) on ViT. This confirms that CAT+ exploits the fundamental vulnerability of the concept-to-label mapping $f(\cdot)$, making it backbone-agnostic.

**Cross-Domain Robustness:** We further validated robustness on the Large-scale Attribute Dataset (LAD), comprising five distinct sub-tasks (Animals, Electronics, Fruits, Vehicles, Hairstyles). As detailed in **Appendix J.2**, CAT+ consistently achieved high ASRs (e.g., 96.69% on LAD-F, 84.56% on LAD-E). This demonstrates that the *Concept Filter* and *Iterative Poisoning* mechanisms are robust to varying data distributions and concept granularities.

## 8 Preliminary of Defense and Results

### 8.1 Analysis of Neural Cleanse Defense Against CAT

In this study, we evaluated the effectiveness of Neural Cleanse in defending against CAT attacks. The backdoored model was configured with the following parameters: CAT Attack, injection rate = 0.1, trigger size = 20, dataset = CUB, and target class = 0. Neural Cleanse is a technique designed to detect and mitigate backdoor attacks in deep learning models by analyzing the model's behavior on specific inputs and identifying potential backdoors.

We applied Neural Cleanse to import the backdoored model and clean test data. Through reverse engineering, we generated trigger and mask images for all categories. Subsequently, we calculated the L1-norm of the reverse-engineered triggers and used this to compute the median, Median Absolute Deviation (MAD), and anomaly index for each category. The anomaly index helps identify potential backdoor target classes by flagging categories with high deviations from the norm.

Due to the large number of classes (200), we present a subset of the results in Table 4.

| Label | L1–norm | Anomaly Index |
|:-----:|:-------:|:-------------:|
| 0 | 25232.596 | 1.038 |
| 1 | 25148.714 | 0.558 |
| 2 | 25489.729 | 2.509 |
| 3 | 25261.745 | 1.205 |
| 5 | 25106.957 | 0.319 |
| 6 | 25182.588 | 0.752 |
| 7 | 24880.898 | 0.975 |
| 8 | 25053.514 | 0.013 |
| 9 | 24892.678 | 0.907 |
| 10 | 24867.008 | 1.054 |
| 11 | 24980.345 | 0.405 |
| 12 | 25042.376 | 0.051 |
| 13 | 25349.604 | 1.707 |
| 16 | 24658.980 | 2.244 |
| 44 | 24596.369 | **2.602** |

Table 4: Anomaly Index for Selected Categories.

Neural Cleanse flagged labels 44 and 16 as anomalies due to their anomaly indices exceeding 2.0. However, this does not align with the true target class, which is label 0. For further analysis, the mask images, and pattern images obtained through reverse engineering for categories 0, 16, and 44 are visualized in Figure 8.

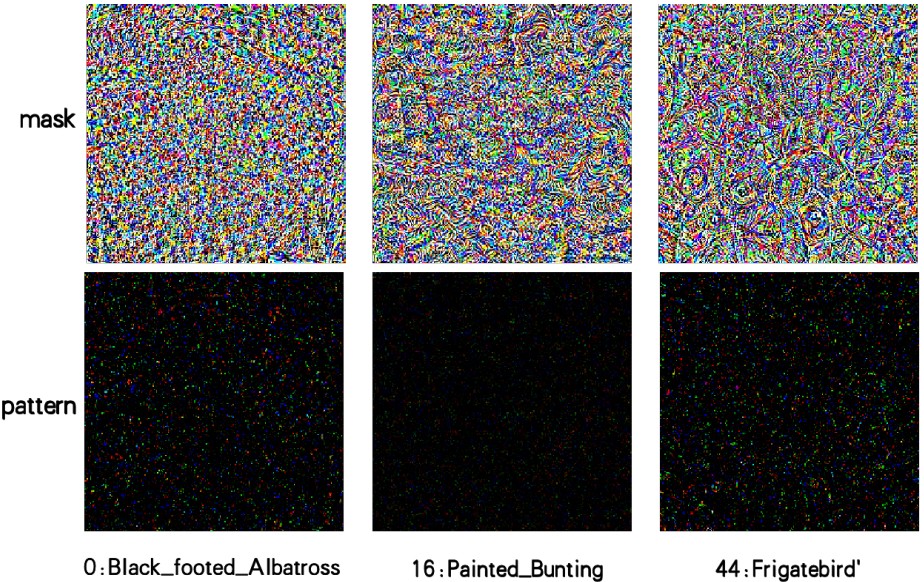

Figure 8: Visualization of Mask Images, and Pattern Images for Classes 0, 16, and 44.

This failure can be attributed to the unique characteristics of CAT attacks. Unlike conventional backdoor attacks that manipulate inputs, outputs, or model structures, CAT targets the concept layer during training. This attack exploits the reliance of CBMs on interpretable representations, making it distinct and challenging to detect using methods designed for more straightforward backdoor attacks. The attacker has access to the training data but lacks direct control over the concept space during inference, leading to subtle and stealthy manipulations that are not easily observable through standard analysis techniques.

These results suggest that Neural Cleanse may not be effective in defending against CAT attacks. This highlights the need for developing specialized defense mechanisms that can address the unique characteristics of concept-level backdoor attacks. Further research is required to better protect CBMs from such sophisticated threats.

**Why Standard Defenses Fail:** As shown in Table 4, Neural Cleanse fails to correctly identify the target class (Label 0). This failure stems from the fundamental difference between pixel-space and concept-space optimization. Standard backdoor defenses often rely on identifying continuous, high-magnitude artifact patterns (gradients) that lead to a specific class. However, CAT+ operates in a *discrete, semantic space*. The "poisoned" concept vector is essentially a specific combination of binary attributes (e.g., $c_{wing\_color} = 1, c_{beak\_red} = 1$). In natural data, rare combinations of attributes occur frequently (e.g., a specific biological mutation). Therefore, the poisoned samples do not appear as statistical outliers in the gradient or latent space, but rather as valid "tail" samples of the data distribution. This effectively bypasses defenses designed for continuous pixel perturbations.

### 8.2 Demo about a Defense Method Which We Design

Given an training dataset $\mathcal{D} = \{(\mathbf{x}_1, \mathbf{c}_1, y_1), (\mathbf{x}_2, \mathbf{c}_2, y_2), \cdots, (\mathbf{x}_n, \mathbf{c}_n, y_n)\}$, where $n$ is the number of data. For concept vectors $\mathbf{c}$, we first encode them from text form into embedding, then use clustering algorithm to cluster them into $m$ groups $\mathcal{F}^j(\mathbf{c}_i)$. Then we divide training dataset into groups following the index of $\mathbf{c}_i$ to generate $m$ sub-datasets. After preparation for the data, we individually train our model upon every sub-dataset and acquire sub-classifier $f^j$. In testing time, every input concept vector divided by the same clustering method into groups and be predicted as a result. At last, the ensemble result is given by the majority vote through $m$ sub-classifiers. Table 5 shows the result of our prototype.

| Clustering Num | Original | CAT | CAT+ | ASR(CAT) | ASR(CAT+) |
|---|---|---|---|---|---|
| Clustering Num 3 | 83.09 | 77.79 | 77.17 | 30.78↓ | 42.75↓ |
| Clustering Num 4 | 83.03 | 78.75 | 78.56 | 11.55↓ | 17.16↓ |
| Clustering Num 5 | 84.24 | 79.51 | 80.76 | 25.95↓ | 16.64↓ |
| Clustering Num 6 | 84.12 | 80.43 | 80.43 | 23.84↓ | 20.12↓ |

Table 5: The Accuracy (%) for each guard model on clean test data for CUB dataset, the Clustering Num denotes a parameter we propose to use in our futher defense framework, the Original denotes to the accuracy when there is no attack. The CAT and CAT+ value refers to the accuracy of defense model and the ASR refers to attack success rate in different models. The experiment settings: injection rate is 5%, trigger size is 20, and the original ASR are 44.66% and 89.68% of CAT and CAT+, respectively.

## 9 Conclusion

Our study demonstrates that the multimodal architecture of Concept Bottleneck Models (CBMs), designed to bridge visual inputs and textual concepts for interpretability, introduces critical vulnerabilities to concept-level backdoor attacks. Through CAT and CAT+, we reveal how subtle manipulations of semantic concepts—often imperceptible to both human auditors and traditional detection mechanisms—can systematically hijack model predictions while preserving clean-data performance. These attacks exploit the inherent sparsity and subjectivity of concept annotations, weaponizing the very interpretability features intended to build trust in AI systems. Our findings necessitate a paradigm shift in securing interpretable AI: security mechanisms must now account for cross-modal threat propagation between visual and conceptual representations, while developing dynamic auditing protocols that monitor concept-semantic consistency. This work serves as both a warning and a roadmap—highlighting the delicate balance between explainability and security in next-generation AI systems, and urging collaborative efforts to harden multimodal reasoning against sophisticated semantic attacks.

## Appendix Introduction

For a comprehensive exploration of CAT, we provide extensive theoretical proofs, detailed explanations, and additional experiments in the Appendix. This supplementary material begins with an analysis of limitations and introduces the novel *Image2Trigger_c* problem in Appendix A, addressing challenges in concept-level trigger activation during testing. Algorithmic implementations are detailed in Appendix B (Algorithms 1-3), supported by rigorous theoretical analyses in Appendices C-D, which establish mathematical foundations for attack effectiveness, performance bounds, and Bayesian probability estimations. The iterative poisoning strategy is further elaborated in Appendix G using information gain metrics. Experimental setups, including datasets (CUB, AwA) in Appendix H, are extended with cross-dataset and backbone evaluations in Appendix J. Human evaluation protocols in Appendix K and LLM-based detection analyses in Appendix L quantify attack stealthiness (Table 22). Appendix M extends the framework to continuous concept spaces, while preliminary Image2Trigger_c visualizations in Appendix ?? (Figure 7) demonstrate its potential. The appendix concludes with ethical considerations and reproducibility commitments.

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

## A   Limitation and Analysis

Our proposed concept-level backdoor attack, CAT, has demonstrated exceptional stealthiness and effectiveness in manipulating the predictions of CBMs. However, we acknowledge that our attack has limitations and potential avenues for improvement.

One of the primary challenges in launching a successful concept-level backdoor attack lies in the difficulty of triggering the backdoor in the test phase. Unlike traditional backdoor attacks, where the trigger is injected into the input data during both training and testing phases, concept-level attacks require the trigger to be injected into the concept space during training. However, during testing, we can only input the image, without direct access to the concept space, but we can still achieve our attack goals by mixing poisonous datasets embedded with triggers into the training set, without the need to directly manipulate the concept layer.

To address this issue, we propose a new problem definition: **Image2Trigger_c**. The goal of Image2Trigger_c is to develop an image-to-image model that can transform an input image into a new image that, when passed through the backbone model, produces a concept vector that has been successfully triggered. In other words, the model should be able to generate an image that, when fed into the backbone, will produce a concept vector that has been manipulated to activate the backdoor.

Formally, we define the Image2Trigger_c problem as follows:

Given an input image $x$, a backbone model $g$, and a target concept vector $\mathbf{c}_{tc}$, the goal is to find an image-to-image model $F$ that can generate an image $x'$ such that $g(x') = \mathbf{c}_{tc}$ where $\mathbf{c}_{tc}$ is the target concept vector that has been manipulated to activate the backdoor.

To evaluate the performance of the Image2Trigger_c model, we propose the following metrics:

**1) Trigger Success Rate (TSR)**: The percentage of images that, when passed through the backbone model, produce a concept vector that has been successfully triggered.

**2) Image Similarity (IS)**: A measure of the similarity between the original image $x$ and the generated image $x'$. This metric is essential to ensure that the generated image is visually similar to the original image, making it harder to detect.

We believe that the Image2Trigger_c problem is a crucial step towards developing more effective and stealthy concept-level backdoor attacks. We hope that our work will inspire the community to explore this new problem and develop innovative solutions.

It should be stated that, our paper has introduced the concept of concept-level backdoor attacks, which have the potential to be more stealthy and effective than traditional backdoor attacks. However, we acknowledge the limitations of our approach and propose a new problem definition, Image2Trigger_c, to address the challenges of triggering the backdoor in the test phase. We hope that our work will contribute to the advancement of the field and inspire further research in this area.

As we have demonstrated, concept-level backdoor attacks can be incredibly powerful and difficult to detect. However, we believe that there is still much to be explored in this area, and we hope that our work will serve as a catalyst for further research. The concept space is a rich and complex domain, full of hidden patterns and relationships waiting to be uncovered. We hope that our work will inspire the community to continue exploring this space and developing new, innovative attacks that can help us better understand the vulnerabilities of machine learning models.

## B  Pseudocode

This appendix provides the pseudocode for the training and testing procedures of CAT and its enhanced version, CAT+ in Algorithm 1, Algorithm 2 and Algorithm 3.

**Algorithm 1**: Train Time CAT

**Input:** Clean dataset $\mathcal{D}$, cache dataset $\mathcal{D}_{cache}$, target label $y_{tc}$, trigger size $|\mathbf{e}|$, injection rate $p$, poisoning function $T_e$
**Output:** Poisoned dataset $\mathcal{D}'$, compromised model $f$

1. Fit a regressor on $\mathcal{D}_{cache}$ to obtain coefficients $\hat{\beta}$.

2. Select $|\mathbf{e}|$ concepts with the smallest $|\hat{\beta}_i|$.

3. Set $\tilde{\mathbf{c}} \leftarrow \mathbf{e}$ based on dataset sparsity.

4. Let $\mathcal{D}_{tc}$ be the subset of $\mathcal{D}$ with label $y_{tc}$.

5. Randomly select $\mathcal{D}_{adv}$ with size $p \times |\mathcal{D}|$ from $\mathcal{D} - \mathcal{D}_{tc}$.

6. Apply $T_e$ to $\mathcal{D}_{adv}$ to obtain $\tilde{\mathcal{D}}_{adv}$.

7. Construct $\mathcal{D}' \leftarrow \mathcal{D} + \tilde{\mathcal{D}}_{adv} - \mathcal{D}_{adv}$.

8. Train $f \leftarrow \mathcal{A}(\mathcal{D}')$.

9. Return $\mathcal{D}'$ and $f$.

**Algorithm 2**: Test Time CAT

**Input:** Retrained CBM $f$, clean test dataset $\mathcal{D}_{\text{test}}$, cache dataset $\mathcal{D}_{cache}$, trigger size $|\mathbf{e}|$, trigger concepts $\mathbf{e}$, trigger $\tilde{\mathbf{c}}$, target class $y_{tc}$
**Output:** $Acc_{(\text{retrained;w/o};T_{\mathbf{e}})}$, $Acc_{(\text{retrained;w/};T_{\mathbf{e}})}$, $\Delta Acc$, $p$

1. Evaluate $f$ on $\mathcal{D}_{test}$ to obtain $Acc_{(\text{retrained;w/o};T_{\mathbf{e}})}$.

2. Remove samples labeled as $y_{tc}$ from $\mathcal{D}_{test}$ to form $\mathcal{D}'_{test}$.

3. Apply $T_e$ to $\mathcal{D}'_{test}$ to obtain $\mathcal{D}_{cache}$.

4. Evaluate $f$ on $\mathcal{D}_{cache}$ to obtain $Acc_{(\text{retrained;w/};T_{\mathbf{e}})}$.

5. Return $Acc_{(\text{retrained;w/o};T_{\mathbf{e}})}$ and $Acc_{(\text{retrained;w/};T_{\mathbf{e}})}$.

**Algorithm 3**: Train Time CAT+

**Input:** Clean dataset $\mathcal{D}$, target label $y_{tc}$, trigger size $|\mathbf{e}|$, concept set $\mathbf{c}$, operation set $P_c$
**Output:** $\mathcal{D}'$, $f$

1. Initialize $\tilde{\mathbf{c}} \leftarrow \emptyset$ and let $p_0$ be the fraction of samples with label $y_{tc}$ in $\mathcal{D}$.

2. While $|\tilde{\mathbf{c}}| < |\mathbf{e}|$, do:

    (a) Set $\mathcal{Z}_{\max} \leftarrow 0$, $c_{\text{select}} \leftarrow \emptyset$, and $P_{\text{select}} \leftarrow \emptyset$.
    (b) For each $c \in \mathbf{c}$ and $P \in P_c$:
        i. Apply operation $P$ to concept $c$ in $\mathcal{D}$ to obtain $c_a$.
        ii. Calculate the conditional probability $p^{(\text{target}|c_a)}$ of $y_{tc}$ in $c_a$.
        iii. Compute $\mathcal{Z}(c, P)$ using Equation (1).
        iv. If $\mathcal{Z}(c, P) > \mathcal{Z}_{\max}$, update $\mathcal{Z}_{\max}$, $c_{\text{select}}$, and $P_{\text{select}}$.
    (c) Update $\tilde{\mathbf{c}} \leftarrow \tilde{\mathbf{c}} \cup \{c_{\text{select}}\}$.
    (d) Apply $P_{\text{select}}$ to concept $c_{\text{select}}$ in $\mathcal{D}$.

3. Inject trigger $\tilde{\mathbf{c}}$ into $\mathcal{D}$ to obtain $\mathcal{D}'$.

4. Train $f \leftarrow \mathcal{A}(\mathcal{D}')$.

5. Return $\mathcal{D}'$ and $f$.

## C  Proof of Minimal Performance Degradation (without Trigger)

To prove that the retrained CBM's performance on the original dataset $\mathcal{D}$ is minimally affected when the trigger is not activated, we consider the following. Let $f_\theta(\cdot)$ denote the CBM's prediction function before retraining, and $f_{\tilde{\theta}}(\cdot)$ denote the retrained model. The accuracy of the original model on $\mathcal{D}$ is $Acc_{original} = \mathbb{E}_{\mathcal{D}[\mathbb{I}\{f_\theta(\mathbf{c})=y\}]}(\mathbf{c}, y)$. Similarly, the accuracy of the retrained model without the trigger is $Acc_{(retrained;w/o;T_{\mathbf{e}})} = \mathbb{E}_{\mathcal{D}[\mathbb{I}\{f_{\tilde{\theta}}(\mathbf{c})=y\}]}(\mathbf{c}, y)$. Assuming that the retraining process does not significantly alter the model's behavior on clean data, we have:

$$
\begin{aligned}
& Acc_{(retrained;w/o;T_{\mathbf{e}})} - Acc_{original} \\
& \approx \mathbb{E}_{\mathcal{D}[\mathbb{I}\{f_{\tilde{\theta}}(\mathbf{c})=y\}]}(\mathbf{c}, y) - \mathbb{E}_{\mathcal{D}[\mathbb{I}\{f_\theta(\mathbf{c})=y\}]}(\mathbf{c}, y) \\
& = \mathbb{E}_{\mathcal{D}[\mathbb{I}\{f_{\tilde{\theta}}(\mathbf{c})\neq f_\theta(\mathbf{c})]\}}(\mathbf{c}, y) \\
& \leq \mathcal{P}\{f_{\tilde{\theta}}(\mathbf{c}) \neq f_\theta(\mathbf{c})\} \\
& = \epsilon,
\end{aligned}
$$

where $\epsilon$ is a small positive value representing the maximum possible difference in accuracy due to retraining. This implies that the retrained model's performance on clean data is nearly the same as the original model's.

# D    Derivation of the Upper Bound for Accuracy Decrease (with Trigger)

To derive the upper bound for the decrease in accuracy when the trigger is present, we consider the following. Let $Acc_{\mathbf{c}_i}$ denote the accuracy of the retrained model on data point $\mathbf{c}_i$ without the trigger, and $Acc_{(\mathbf{c}_i;T_\mathbf{e})}$ denote the accuracy with the trigger. We have:

$$
\begin{aligned}
& Acc_{(retrained;w/o;T_\mathbf{e})} - Acc_{(retrained;w/;T_\mathbf{e})} \\
&= \mathbb{E}_{(\mathbf{c}_i,y_i)\sim\mathcal{D}_{test}}[Acc_{\mathbf{c}_i}] - \mathbb{E}_{(\mathbf{c}_i,y_i)\sim\mathcal{D}_{test}}[Acc_{(\mathbf{c}_i;T_\mathbf{e})}] \\
&= \mathbb{E}_{(\mathbf{c}_i,y_i)\sim\mathcal{D}_{test}}[Acc_{\mathbf{c}_i} - Acc_{(\mathbf{c}_i;T_\mathbf{e})}] \\
&\leq \mathbb{E}_{(\mathbf{c}_i,y_i)\sim\mathcal{D}_{test}}[\Delta Acc] \\
&= p \cdot \Delta Acc,
\end{aligned}
$$

where $p$ is the fraction of data points with the trigger, and $\Delta Acc = \mathbb{E}_{(\mathbf{c}_i,y_i)\sim\mathcal{D}_{test}}[Acc_{(\mathbf{c}_i;T_\mathbf{e})} - Acc_{\mathbf{c}_i}]$ is the average decrease in accuracy due to the trigger. This upper bound indicates that the decrease in accuracy is directly proportional to the fraction of data points with the trigger.

In this section, we derive the lower and upper bounds for the success rate of the CAT attack during the Test Time II phase. Let $Acc_{(retrained;w/o;T_\mathbf{e})}$ denote the accuracy of the retrained CBM on clean data without the trigger, and $Acc_{(retrained;w/;T_\mathbf{e})}$ be the accuracy on data with the trigger activated. The success rate of the attack is defined as the proportion of instances originally not labeled as $y_{tc}$ that are misclassified as $y_{tc}$ due to the trigger. Assuming that the trigger is effective, we expect a significant decrease in accuracy when the trigger is present. Let $\Delta Acc$ denote the difference between the accuracy with and without the trigger:

$$
\Delta Acc = Acc_{(retrained;w/o;T_\mathbf{e})} - Acc_{(retrained;w/;T_\mathbf{e})}. \tag{7}
$$

The lower bound for the success rate can be derived as:

$$
\text{Success Rate} \geq \frac{\Delta Acc}{1 - Acc_{(retrained;w/o;T_\mathbf{e})}}. \tag{8}
$$

This bound represents the minimum success rate achievable if all the misclassifications due to the trigger are instances originally not labeled as $y_{tc}$. Conversely, the upper bound for the success rate is given by:

$$
\text{Success Rate} \leq \frac{\Delta Acc}{Acc_{(retrained;w/o;T_\mathbf{e})}}. \tag{9}
$$

This bound assumes that all the misclassifications due to the trigger are from instances originally not labeled as $y_{tc}$, and no clean instances are misclassified. By evaluating the model's performance within these bounds, we can assess the effectiveness of the CAT attack in practice.

# E    Lower or Upper Bound for Bayesian CAT

We will employ Bayesian methods to estimate the probability of trigger activated in CAT, and use this to optimize our experimental attempts in further sections. We assume that $\theta$ is the probability of trigger activated in CAT and $\theta \in [0,1]$. Assuming that we have conducted $N$ backdoor injection experiments on the dataset and the backdoor was triggered $k$ times, where $N$ and $k$ are given. Now we will derive the prior distribution for $\theta$. Clearly, the activation of the trigger will result in one of two states: 1 or 0. The Beta distribution is defined over the interval $[0,1]$ and is a conjugate prior for the binomial distribution, allowing us to obtain a closed-form solution. Therefore, we will use the Beta distribution here, i.e., $\theta \sim \text{Beta}(\alpha, \beta)$. Note that the parameter $\beta$ here is different from the regressor $f$ ones. Then the PDF(prior probability density function) for $\theta$ using the Beta distribution can be expressed as follows:

$$
p(\theta) = \frac{\Gamma(\alpha + \beta)}{\Gamma(\alpha)\Gamma(\beta)}\theta^{\alpha-1}(1-\theta)^{\beta-1},\ 0 \leq \theta \leq 1, \tag{10}
$$

where $\alpha, \beta$ are the prior parameters, $\Gamma(\cdot)$ is the Gamma function.

Now we will establish the likelihood function for the parameter $\theta$. We assume that the probability of triggering a backdoor in each backdoor injection experiment is independent, and through observation, $k$ out of $N$ experiments are successful. The likelihood function for a binomial distribution is:

$$L(\theta) = p(k|\theta) = \binom{N}{k} \theta^k (1-\theta)^{N-k} \tag{11}$$

According to Bayes' theorem, we obtain:

$$p(\theta|k) = \frac{L(\theta)p(\theta)}{\int_0^1 L(\theta)p(\theta) \, d\theta}, \tag{12}$$

where the denominator is a normalization constant and is independent of $\theta$, we can first calculate the unnormalized posterior distribution and then normalize it by identifying its distribution form.

The unnormalized posterior distribution will satisfy:

$$p(\theta|k) \propto \theta^k(1-\theta)^{N-k} \cdot \theta^{\alpha-1}(1-\theta)^{\beta-1}$$
$$= \theta^{\alpha+k-1}(1-\theta)^{\beta+N-k-1}$$

Same with Beta distribution, this distribution form is identified as:

$$\theta|k \sim \text{Beta}(\alpha', \beta'), \tag{13}$$

where posterior parameters $\alpha' = \alpha + k, \beta' = \beta + N - k$.

Using the posterior distribution we can derive the upper and lower bound of our CAT. Our goal is to obtain the $(1-\gamma)\%$ confidence interval for $\theta$. Recall the definition of lower or upper bound:

$$p(\theta \le \theta_{lower}) = \frac{\gamma}{2}, \quad p(\theta \le \theta_{upper}) = 1 - \frac{\gamma}{2} \tag{14}$$

The bounds for $\theta$ could be expressed below:

$$\theta_{lower} = \text{BetaCDF}^{-1}(\frac{\gamma}{2}, \alpha', \beta') \tag{15}$$

$$\theta_{upper} = \text{BetaCDF}^{-1}(1 - \frac{\gamma}{2}, \alpha', \beta'), \tag{16}$$

where the term $\text{BetaCDF}^{-1}(p, \alpha', \beta')$ represents the $p$-th quantile of the Beta distribution with parameters $\alpha'$ and $\beta'$, and CDF is the cumulative distribution function.

### E.1 Parameter Estimation

The PDF of beta distribution is formed as:

$$f(\theta; \alpha, \beta) = \frac{\theta^{\alpha-1}(1-\theta)^{\beta-1}}{B(\alpha, \beta)}, \tag{17}$$

where term $B(\alpha, \beta)$ is the Beta function. We use MLE (Maximum Likelihood Estimation) to estimate the parameter in beta distribution. Assuming that the observations value of $\theta$ are $\{\theta_1, \theta_2, \cdots, \theta_n\}$ and $\theta_i \in [0, 1]$. The likelihood function of Beta distribution is expressed as:

$$L(\alpha, \beta) = \prod_{i=1}^{n} f(\theta_i, \alpha, \beta), \tag{18}$$

and we transform it into logarithm format:

$$\log L(\alpha, \beta) = \Sigma_{i=1}^{n}[(\alpha - 1)\log(\theta_i) + (\beta - 1)\log(1 - \theta_i)] - n \log B(\alpha, \beta), \tag{19}$$

and the Beta function could be calculated by Gamma function:

$$B(\alpha, \beta) = \frac{\Gamma(\alpha)\Gamma(\beta)}{\Gamma(\alpha + \beta)}.$$

Finally we solve the optimal problems with parameter $\alpha, \beta$ to meet the requirement:

$$\max \log L(\alpha, \beta) = \max \Sigma_{i=1}^{n}[(\alpha - 1)\log(\theta_i) + (\beta - 1)\log(1 - \theta_i)] - n \log B(\alpha, \beta).$$

## F  CAT Robustness

In our CAT framework, we will not only consider the effectiveness of the attack but also evaluate its robustness against generalized defenses. Models based on random perturbations tend to have strong generalization capabilities, and we will derive the probability of activating the trigger even under random perturbations. Assume that $S \in \mathcal{S}$ is a random perturbation from perturbation space, the definition of CAT Robustness could be expressed as below:

$$R = P_{\mathbf{c}, S}\{f(S(\mathbf{c} \oplus \tilde{\mathbf{c}})) = y_{tc}\}, \tag{20}$$

where term $P_{\mathbf{c}, S}$ represents the joint probability distribution of $\mathbf{c}$ and $S$. In CAT, concept vector $\mathbf{c}$ and perturbation $S$ are random independent variables. So we could decompose the joint probability distribution into the following expression:

$$P_{\mathbf{c}, S} = P_{\mathbf{c}} \cdot P_S, \tag{21}$$

then the CAT robustness could be expressed as follows:

$$R = \int_{\mathcal{C}} \int_{\mathcal{S}} \mathbb{I}\{f(S(\mathbf{c} \oplus \tilde{\mathbf{c}})) = y_{tc}\} dP_{\mathbf{c}}(\mathbf{c}) dP_S(S) \tag{22}$$

where equation 22 is a stochastic differential equation. For a fixed concept vector $\mathbf{c}$, the CAT robustness will be follows:

$$R_{\mathbf{c}} = P_S\{f(S(\mathbf{c} \oplus \tilde{\mathbf{c}})) = y_{tc}|\mathbf{c}\} \tag{23}$$

Therefore, the overall CAT robustness is:

$$R = \mathbb{E}_{\mathbf{c}}[R_{\mathbf{c}}] = \int_{\mathcal{C}} R_{\mathbf{c}} dP_{\mathbf{c}}(\mathbf{c}) \tag{24}$$

## G  Theoretical Foundation of Iterative Poisoning

The iterative poisoning strategy in CAT+ is grounded in the concept of maximizing the impact of the backdoor trigger while maintaining stealthiness. To formalize this, we first introduce the concept of *information gain* to quantify the change in the model's understanding of the target class after applying the trigger.

**Information Gain:** The information gain $\mathcal{I}(c_{select}, P_{select})$ is a measure of the additional information the model gains about the target class $y_{tc}$ when the concept $c_{select}$ is perturbed using operation $P_{select}$. It can be defined as the mutual information between the target class and the perturbed concept, given by:

$$\mathcal{I}(c_{select}, P_{select}) = \mathbb{H}(y_{tc}) - \mathbb{H}(y_{tc}|c_{select}, P_{select}), \tag{25}$$

where $\mathbb{H}(y_{tc})$ is the entropy of the target class distribution and $\mathbb{H}(y_{tc}|c_{select}, P_{select})$ is the conditional entropy of the target class given the perturbed concept.

**Optimal Concept Selection:** In each iteration, we aim to maximize the information gain to ensure that the trigger has the most significant impact on the model's prediction. To achieve this, we define the *information gain ratio* as:

$$\mathcal{R}(c_{select}, P_{select}) = \frac{\mathcal{I}(c_{select}, P_{select})}{\mathbb{H}(y_{tc})}, \tag{26}$$

which represents the relative increase in information about the target class due to the perturbation.

**Z-score Revisited:** The Z-score $\mathcal{Z}(c_{select}, P_{select})$ introduced earlier is closely related to the information gain ratio. In fact, we can show that the Z-score is a monotonic function of the information gain ratio, such that a higher Z-score corresponds to a higher information gain ratio. This relationship allows us to use the Z-score as a proxy for selecting the optimal concept and operation in each iteration.

## H   Datasets

**CUB.** The Caltech-UCSD Birds-200-2011 (CUB)Wah et al. (2011) dataset is designed for bird classification and contains 11,788 bird photographs across 200 species. Additionally, it includes 312 binary bird attributes to represent high-level semantic information. Previous work mostly followed the preprocessing steps outlined by Koh et al. (2020). They first applied majority voting to resolve concept disparities across instances from the same class, then selected attributes that appeared in at least 10 classes, ultimately narrowing the selection to 116 binary attributes. However, in our experiments, we preprocess the data slightly differently. We do not attempt to eliminate the disparity across instances from the same class, meaning we accept that instances from the same species may have different concept representations. Additionally, we select high-frequency attributes at the instance level—specifically, we only use attributes that appear in at least 500 instances. Finally, we retain 116 attributes, with over 90% overlap with the attributes selected by Koh et al. (2020).

**AwA.** The Animals with Attributes (AwA) Xian et al. (2018) dataset contains 37,322 images across 50 animal categories, with each image annotated with 85 binary attributes. We split the images equally by class into training and test datasets, resulting in 18,652 images in the training set and 18,670 images in the test set. There is no modification for the binary attributes.

## I   Extend Experiment Results

In this section, we give more detailed experiment results, and we also conducted our attack experiments across different target classes. Table 6 and Table 7 gives the detailed experiment results for CUB dataset when the target class is set to 0, we see that CAT+ can achieve high ASR, while maintain a high performance on clean data, showing the effectiveness of CAT+, even with a small injection rate (1%). Table 8 and Table 9 shows the detailed experiment results for AwA dataset when the target class is set to 0, while Table 10 and Table 11 shows the detailed experiment results for AwA dataset when the target class is set to 2. These results shows that our CAT and CAT+ works well across different target classes. Table 12 gives the results on CUB dataset with a fixed trigger size and injection rate, which shows some performance disparity across different target classes.

## J   More experiments

### J.1   Experiments on Different Backbones

We evaluate the attack performance by using another pre-trained vision backbone, Vision Transformer (VIT)Dosovitskiy (2020), we resize the input images to $224 \times 224$ to fit the input dimension of VIT, the results are shown in Table 14. When the pre-trained backbone comes to VIT, our CAT+ still keeps both the stealthiness and high ASR: The lowest task accuracy with no trigger activated is 81.48%, which guarantees the stealthiness of our attack, and highest ASR comes to 72.05%. This experiment claims that our CAT+ adapts different pre-trained backbone and portability.

|  | 1% | | 2% | | 3% | | 5% | | 7% | | 10% | |
|---|---|---|---|---|---|---|---|---|---|---|---|---|
|  | CAT | CAT+ | CAT | CAT+ | CAT | CAT+ | CAT | CAT+ | CAT | CAT+ | CAT | CAT+ |
| 2 | 0.52 | 3.85 | 1.27 | 8.50 | 1.94 | 9.58 | 2.85 | 14.28 | 4.49 | 20.44 | 7.55 | 26.65 |
| 5 | 0.23 | 12.11 | 0.92 | 19.99 | 1.51 | 24.58 | 2.01 | 28.40 | 4.37 | 40.42 | 7.15 | 42.45 |
| 8 | 2.06 | 23.09 | 5.12 | 36.80 | 5.97 | 43.15 | 7.46 | 44.26 | 14.69 | 55.38 | 19.17 | 63.72 |
| 10 | 0.90 | 18.06 | 3.57 | 26.63 | 6.11 | 42.21 | 8.34 | 47.33 | 9.39 | 52.38 | 16.20 | 50.78 |
| 12 | 3.85 | 19.08 | 11.54 | 37.23 | 18.20 | 49.84 | 17.80 | 51.58 | 28.19 | 56.52 | 36.26 | 64.40 |
| 15 | 3.92 | 14.10 | 13.58 | 24.08 | 16.78 | 31.80 | 22.95 | 32.43 | 29.56 | 38.67 | 37.46 | 39.37 |
| 17 | 8.26 | 22.80 | 23.18 | 35.69 | 29.27 | 47.76 | 39.56 | 54.70 | 48.32 | 63.60 | 52.64 | 66.69 |
| 20 | 6.89 | 57.70 | 24.36 | 77.65 | 30.99 | 84.47 | 35.90 | 87.51 | 48.77 | 90.16 | 59.28 | 93.01 |
| 23 | 6.04 | 74.76 | 18.95 | 84.00 | 25.45 | 87.82 | 26.32 | 91.92 | 43.36 | 87.20 | 52.41 | 90.65 |

Table 6: ASR(%) under different injection rates (1% – 10%) and trigger size (2 – 23) in CUB dataset, target class 0.

|  | 1% | | 2% | | 3% | | 5% | | 7% | | 10% | |
|---|---|---|---|---|---|---|---|---|---|---|---|---|
|  | CAT | CAT+ | CAT | CAT+ | CAT | CAT+ | CAT | CAT+ | CAT | CAT+ | CAT | CAT+ |
| 2 | 80.58 | 80.70 | 80.08 | 80.05 | 79.08 | 79.72 | 78.75 | 78.51 | 76.99 | 77.55 | 74.56 | 74.27 |
| 5 | 80.81 | 80.00 | 79.92 | 79.72 | 80.12 | 79.50 | 78.44 | 78.46 | 76.94 | 76.46 | 78.92 | 74.68 |
| 8 | 80.95 | 80.74 | 79.74 | 79.75 | 79.50 | 79.72 | 77.94 | 78.49 | 77.11 | 77.55 | 74.84 | 75.13 |
| 10 | 80.70 | 80.43 | 80.27 | 80.19 | 79.51 | 79.38 | 78.31 | 78.31 | 77.34 | 77.06 | 74.82 | 72.95 |
| 12 | 81.08 | 80.62 | 79.93 | 79.82 | 79.03 | 79.22 | 78.55 | 77.67 | 77.42 | 76.91 | 74.99 | 74.85 |
| 15 | 80.72 | 80.24 | 80.19 | 79.88 | 80.00 | 78.68 | 78.34 | 77.49 | 77.46 | 77.08 | 74.96 | 74.91 |
| 17 | 80.84 | 80.51 | 79.82 | 79.96 | 79.20 | 79.58 | 78.31 | 78.29 | 76.84 | 77.32 | 75.53 | 74.91 |
| 20 | 80.82 | 80.81 | 80.39 | 80.26 | 79.70 | 79.93 | 78.22 | 78.82 | 77.46 | 77.49 | 75.03 | 75.53 |
| 23 | 80.57 | 80.19 | 80.31 | 80.43 | 79.58 | 79.62 | 78.72 | 78.10 | 77.72 | 76.96 | 75.42 | 74.83 |

Table 7: Task Accuracy under different injection rates (1% – 10%) and trigger size (2 – 23) of test in CUB dataset, target class 0. Task Accuracy shows the effectiveness of mapping $f$ in our CATs. The original task accuracy in CUB is 80.70%.

## J.2 Experiments on More Datasets

We follow the same experiment settings in Section 7 and evaluate the attack performance in each sub-datasets except for LAD-H, for the original accuracy of LAD-H is not ideal. The original accuracies for each sub-datasets are shown in Table 15. We evaluate the performances of CAT and CAT+ on LAD-A and LAD-E, the results are shown in Table 16 17, 18, 19, and we evaluate the performance of CAT on LAD-F and CAT+ on LAD-V, the results are shown in 20, 21. By evaluating each sub-dataset, we observed a significant increase in attack success rates (ASR) as the injection rate increased, particularly with the CAT+ method, which demonstrated more pronounced effectiveness compared to the CAT method.

On the LAD-E dataset, where the original accuracy was 77.82%, both CAT and CAT+ showed similar trends. Despite the increased challenge posed by the LAD-E dataset, the CAT+ method still achieved a high ASR, particularly at a 10% injection rate and larger trigger sizes. In this case, CAT+ achieved an ASR of 84.56%, compared to 50.33% for the CAT method, further underscoring the superior performance of CAT+.

For the LAD-F dataset, with an original accuracy of 89.59%, we found that even at low injection rates (such as 2%), the CAT+ method exhibited a high ASR, reaching as high as 96.69%. This result further validates the broad applicability and strong effectiveness of the CAT+ method across various tasks and datasets.

On the LAD-V dataset, the performance of both the CAT and CAT+ methods followed a similar pattern, but the CAT+ method consistently achieved higher ASR, particularly at higher injection rates, where the ASR reached over 80%. This indicates that CAT+ performs especially well on this dataset.

|  | 2% | | 5% | | 10% | |
|---|---|---|---|---|---|---|
|  | CAT | CAT+ | CAT | CAT+ | CAT | CAT+ |
| 2 | 3.14 | 0.50 | 7.31 | 1.54 | 15.18 | 4.29 |
| 5 | 5.87 | 0.48 | 13.71 | 1.51 | 27.67 | 5.39 |
| 8 | 9.22 | 1.74 | 23.35 | 5.95 | 41.17 | 12.02 |
| 10 | 14.26 | 4.48 | 26.61 | 29.70 | 44.57 | 47.20 |
| 12 | 24.96 | 1.13 | 41.11 | 41.49 | 55.17 | 59.95 |
| 15 | 26.89 | 2.53 | 42.30 | 5.77 | 59.49 | 62.74 |
| 17 | 31.43 | 25.32 | 50.03 | 45.37 | 64.80 | 65.32 |

Table 8: ASR (%) under different injection rates (2%, 5%, 10%) and trigger size $(2 - 17)$ in AwA dataset, target class 0.

|  | 2% | | 5% | | 10% | |
|---|---|---|---|---|---|---|
|  | CAT | CAT+ | CAT | CAT+ | CAT | CAT+ |
| 2 | 83.06 | 83.09 | 81.20 | 81.29 | 78.48 | 77.04 |
| 5 | 83.12 | 83.18 | 80.88 | 81.20 | 76.59 | 76.53 |
| 8 | 83.02 | 83.19 | 80.67 | 80.87 | 76.35 | 76.85 |
| 10 | 83.23 | 83.20 | 80.86 | 81.11 | 78.64 | 72.45 |
| 12 | 83.36 | 82.79 | 80.70 | 80.76 | 76.90 | 76.68 |
| 15 | 83.37 | 83.37 | 80.86 | 80.93 | 76.40 | 76.56 |
| 17 | 83.00 | 82.87 | 80.87 | 80.62 | 76.13 | 76.99 |

Table 9: Task Accuracy under different injection rates (2%, 5%, 10%) and trigger size $(2 - 17)$ in AwA dataset, target class 0. Task Accuracy shows the effectiveness of mapping $f$ in our CATs. The original task accuracy for AwA is 84.68%.

Overall, the CAT+ method consistently demonstrated a significant advantage in attack success rate across most sub-datasets. These results confirm the superior effectiveness of the CAT+ method, especially in multi-task learning scenarios with high injection rates and larger trigger sizes. In comparison, although the CAT method also achieved relatively high attack success rates in some cases, the CAT+ method's overall superiority across multiple datasets was more pronounced.

## K    Human Evaluation Details

### K.1    Human Evaluation Protocol

The human evaluators were provided with the following instructions:

1. **Dataset Description:** You will be presented with a dataset consisting of 60 concept representations, each associated with an input sample (x) and its corresponding class labels (c, y). Among these, 30 concept representations have been backdoored using a concept-based trigger, while the remaining 30 are clean.

2. **Task:** Your task is to identify which data has been backdoored.

3. **Evaluation Criteria:** Analyze the concept space for any subtle modifications that might indicate the presence of a backdoor trigger. Avoid relying on the input samples or class labels.

### K.2    Post-Evaluation Interviews and Insights

After completing the evaluation, the evaluators were interviewed to gather their thoughts and insights on the task. The interview questions included:

|     | 2% | | 5% | | 10% | |
| --- | --- | --- | --- | --- | --- | --- |
|     | CAT | CAT+ | CAT | CAT+ | CAT | CAT+ |
| 2   | 2.27  | 2.09  | 3.18  | 4.55  | 10.35 | 13.46 |
| 5   | 5.40  | 5.33  | 12.25 | 12.28 | 22.03 | 24.31 |
| 8   | 13.13 | 13.69 | 20.99 | 24.59 | 39.67 | 44.93 |
| 10  | 14.32 | 12.76 | 25.17 | 27.02 | 43.79 | 47.56 |
| 12  | 21.22 | 19.68 | 29.12 | 51.31 | 49.15 | 58.78 |
| 15  | 34.95 | 40.08 | 51.45 | 56.88 | 74.22 | 75.21 |
| 17  | 44.34 | 44.18 | 62.42 | 64.85 | 79.22 | 79.13 |

Table 10: ASR (%) under different injection rates (2% – 10%) and trigger size (2 – 17) in AwA dataset, target class 2.

|     | 2% | | 5% | | 10% | |
| --- | --- | --- | --- | --- | --- | --- |
|     | CAT | CAT+ | CAT | CAT+ | CAT | CAT+ |
| 2   | 82.79 | 82.76 | 81.51 | 79.86 | 76.89 | 74.44 |
| 5   | 82.91 | 82.81 | 81.26 | 78.81 | 76.83 | 74.49 |
| 8   | 83.00 | 82.64 | 81.47 | 80.03 | 77.03 | 75.11 |
| 10  | 83.42 | 82.59 | 81.81 | 79.59 | 77.16 | 73.75 |
| 12  | 83.20 | 82.21 | 81.03 | 74.07 | 77.41 | 74.32 |
| 15  | 83.15 | 82.28 | 81.42 | 79.35 | 76.61 | 74.86 |
| 17  | 83.26 | 82.51 | 81.01 | 78.65 | 76.61 | 74.32 |

Table 11: Task Accuracy (%) under different injection rates (2% – 10%) and trigger size (2 – 17) in AwA dataset, target class 2. The original task accuracy for AwA is 84.68%.

1. **Q1:** Describe your approach to distinguishing between backdoored and clean concept representations.

2. **Q2:** Did you notice any specific patterns or changes in the concept space that helped you identify the backdoor samples?

3. **Q3:** How difficult was it to identify the backdoor attacks compared to your initial expectations?

4. **Q4:** What factors do you think contributed to the difficulty in detecting the backdoor in the concept space?

**Evaluator 1:**

1. **A1:** I focused on the relationships between concepts and images.

2. **A2:** The trigger concepts seemed to have a more pronounced effect, but no consistent pattern was evident.

3. **A3:** It was much more challenging than expected due to the subtlety of the changes.

4. **A4:** There were so many concepts that I got distracted, and a lot of datasets were actually mislabeled. It felt like looking for a needle in a haystack. It was so painful. **It's like pouring Coca-Cola into Pepsi.**

**Evaluator 2:**

1. **A1:** I looked for inconsistencies or anomalies that didn't align with the expected concept representation.

2. **A2:** There were slight shifts in emphasis on certain concepts, but no clear pattern.

| Target Class | Task Accuracy(%) CAT | CAT+ | ASR(%) CAT | CAT+ |
|---|---|---|---|---|
| 0 | 75.03 | 75.53 | 59.28 | 93.01 |
| 4 | 74.54 | 74.73 | 1.85 | 0.10 |
| 8 | 75.06 | 75.56 | 74.24 | 52.63 |
| 12 | 74.94 | 75.72 | 53.38 | 1.08 |
| 16 | 75.16 | 75.72 | 40.91 | 68.81 |
| 20 | 75.34 | 75.58 | 1.28 | 12.02 |
| 24 | 74.37 | 74.91 | 0.52 | 54.48 |
| 28 | 74.27 | 74.65 | 35.48 | 17.87 |
| 32 | 74.70 | 75.58 | 37.68 | 2.32 |
| 36 | 74.96 | 75.27 | 37.99 | 6.50 |
| 40 | 74.46 | 74.96 | 42.35 | 10.40 |
| 44 | 74.89 | 75.22 | 17.10 | 23.77 |
| 48 | 75.09 | 75.73 | 49.77 | 4.51 |
| 52 | 74.68 | 74.97 | 82.15 | 95.11 |
| 56 | 75.22 | 75.23 | 70.99 | 57.36 |
| 60 | 74.97 | 75.37 | 2.60 | 24.01 |
| 64 | 74.85 | 74.58 | 43.63 | 84.27 |
| 68 | 75.09 | 75.58 | 39.30 | 9.63 |
| 72 | 74.99 | 75.34 | 47.99 | 59.06 |
| 76 | 74.53 | 74.06 | 46.16 | 30.55 |
| 80 | 75.03 | 75.61 | 62.51 | 3.71 |
| 84 | 74.49 | 74.87 | 9.82 | 75.83 |
| 88 | 74.75 | 74.66 | 51.13 | 32.44 |
| 92 | 75.34 | 75.60 | 39.68 | 72.71 |
| 96 | 74.82 | 75.20 | 17.73 | 11.64 |

Table 12: Task Accuracy and ASR for different Target Classes from 0 to 196. The test dataset is CUB, trigger size is 20 and the injection rate is 10% (1 of 2).

| Target Class | Task Accuracy(%) CAT | CAT+ | ASR(%) CAT | CAT+ |
|---|---|---|---|---|
| 100 | 74.94 | 75.42 | 48.96 | 62.16 |
| 104 | 74.84 | 74.92 | 1.07 | 0.07 |
| 108 | 74.49 | 74.63 | 53.02 | 11.00 |
| 112 | 73.92 | 74.46 | 10.76 | 40.42 |
| 116 | 75.35 | 75.51 | 11.50 | 18.83 |
| 120 | 74.58 | 74.59 | 11.49 | 12.58 |
| 124 | 74.97 | 75.70 | 0.24 | 61.04 |
| 128 | 74.66 | 75.15 | 10.15 | 54.34 |
| 132 | 74.99 | 75.66 | 9.54 | 14.23 |
| 136 | 74.73 | 74.99 | 2.95 | 0.83 |
| 140 | 74.58 | 74.97 | 7.86 | 64.01 |
| 144 | 75.28 | 75.35 | 90.61 | 9.94 |
| 148 | 74.78 | 75.42 | 0.36 | 76.79 |
| 152 | 75.09 | 75.22 | 80.46 | 17.19 |
| 156 | 74.54 | 75.94 | 1.37 | 35.98 |
| 160 | 74.99 | 75.16 | 1.80 | 20.07 |
| 164 | 74.85 | 75.56 | 13.78 | 6.61 |
| 168 | 75.11 | 75.83 | 50.44 | 50.70 |
| 172 | 74.70 | 74.73 | 29.96 | 66.69 |
| 176 | 74.77 | 74.92 | 2.29 | 0.03 |
| 180 | 74.91 | 74.58 | 4.35 | 76.96 |
| 184 | 75.16 | 75.66 | 27.93 | 5.95 |
| 188 | 74.44 | 75.73 | 57.41 | 21.74 |
| 192 | 74.25 | 75.75 | 41.39 | 46.30 |
| 196 | 74.73 | 75.80 | 23.72 | 20.96 |

Table 13: Task Accuracy and ASR for different Target Classes from 0 to 196. The test dataset is CUB, trigger size is 20 and the injection rate is 10% (2 of 2).

| | 2% ACC(%) | ASR(%) | 5% ACC(%) | ASR(%) | 10% ACC(%) | ASR(%) |
|---|---|---|---|---|---|---|
| 8 | 86.54 | 9.89 | 85.42 | 16.12 | 83.02 | 20.23 |
| 10 | 86.24 | 7.41 | 85.23 | 13.60 | 82.79 | 22.28 |
| 12 | 86.57 | 11.31 | 85.52 | 21.56 | 83.03 | 31.75 |
| 15 | 86.76 | 20.96 | 85.26 | 30.76 | 82.62 | 42.30 |
| 17 | 86.66 | 43.86 | 85.31 | 58.48 | 82.10 | 70.85 |
| 20 | 86.40 | 48.47 | 84.62 | 60.27 | **81.48** | **72.05** |

Table 14: Task Accuracy(%) and ASR(%) under different injection rates(2% - 10%) and trigger size in CUB dataset, target class 0, vision backbone is a pretrained VIT, the attack mode is fixed to CAT+, the Original Accuracy is 87.30%.

3. **A3:** It was harder than anticipated due to the almost imperceptible changes.

4. **A4:** The subtlety of the trigger and the high dimensionality of the concept space made it challenging.

**Evaluator 3:**

|  | Training Size | Test Size | # of Concept | # of Class | Original ACC(%) |
|---|---|---|---|---|---|
| LAD-A | 9280 | 3960 | 123 | 50 | 88.54 |
| LAD-E | 12916 | 5555 | 75 | 50 | 77.82 |
| LAD-F | 13606 | 5850 | 58 | 50 | 89.59 |
| LAD-V | 11979 | 5101 | 81 | 50 | 84.30 |
| LAD-H | 6829 | 2941 | 22 | 30 | 58.25 |

Table 15: Statistics and Original Task Accuracy(%)of each LAD sub-datasets

|  | 2% | | 5% | | 10% | |
|---|---|---|---|---|---|---|
|  | ACC(%) | ASR(%) | ACC(%) | ASR(%) | ACC(%) | ASR(%) |
| 2 | 87.95 | 8.13 | 85.68 | 10.35 | 81.94 | 21.30 |
| 5 | 87.65 | 33.04 | 85.81 | 30.14 | 81.67 | 33.40 |
| 8 | 87.80 | 55.51 | 85.93 | 57.29 | 81.84 | 63.09 |
| 10 | 87.78 | 61.42 | 85.96 | 60.51 | 82.42 | 65.87 |
| 12 | 87.58 | 50.91 | 85.76 | 54.47 | 81.84 | 62.26 |
| 15 | 87.47 | 55.54 | 85.76 | 60.12 | 82.40 | 61.58 |
| 17 | 88.16 | 60.25 | 86.09 | 61.32 | 82.42 | 65.21 |
| 20 | 87.95 | 68.66 | 85.73 | 74.33 | 81.92 | 14.45 |

Table 16: Task Accuracy(%) and ASR(%) under different injection rates(2% - 10%) and trigger size (2-20) in LAD-A dataset, target class 0, the attack mode is CAT, the Original Accuracy is 88.54%.

1. **A1:** I searched for deviations from expected concept co-occurrence patterns and structure.

2. **A2:** Minor disruptions in co-occurrence patterns were observed, but not consistent enough.

3. **A3:** It was significantly more difficult than expected due to the stealthiness of the changes.

4. **A4:** The trigger's stealthiness and the complexity of the concept space made detection difficult.

## L  LLM Evaluation Details

The LLM evaluation protocol involved the following steps:

1. **Dataset Preparation:** The 60 concept representations (30 backdoor-attacked and 30 clean), along with their corresponding input samples (x) and class labels (c, y), were provided as input to GPT4-Vision.

2. **Prompt Design:** The prompt for GPT4-Vision was, "image This is the concept of this image what i give: concept: weight, determine if it has been poisioneed. If poisioned, output 1; otherwise, output 0."

## M  CAT+ Continuously Extension

Following the same notation from CAT+ in previous discussion, and assuming that there are $c_{num}$ types of values in the concept, the continuously CAT+ function $\mathcal{Z}(\cdot)$ is defined as follows:

(i) Let $n$ be the total number of training samples, and $n_{target}$ be the number of samples from the target class. The initial probability of the target class is $p_0 = n_{target}/n$.

(ii) Given a modified dataset $c_a = \mathcal{D}; c_{select}; P_{select}$, we calculate the conditional probability of the target class given $c_a$ as $p^{(target|c_a)} = \mathbb{H}(target(c_a))/\mathbb{H}(c_a)$, where $\mathbb{H}$ is a function that computes the overall distribution of labels in the dataset.

| | 2% | | 5% | | 10% | |
|---|---|---|---|---|---|---|
| | ACC(%) | ASR(%) | ACC(%) | ASR(%) | ACC(%) | ASR(%) |
| 2 | 87.22 | 14.53 | 85.45 | 19.76 | 83.41 | 32.54 |
| 5 | 88.16 | 12.86 | 86.46 | 31.23 | 83.16 | 37.69 |
| 8 | 88.38 | 37.45 | 85.66 | 43.26 | 83.06 | 52.48 |
| 10 | 87.47 | 35.26 | 86.16 | 48.38 | 82.78 | 64.72 |
| 12 | 87.65 | 53.92 | 85.48 | 69.29 | 81.92 | 69.24 |
| 15 | 87.90 | 64.56 | 86.16 | 67.69 | 82.42 | 79.67 |
| 17 | 87.73 | 45.19 | 85.58 | 63.07 | 83.08 | 74.93 |
| 20 | 87.20 | 53.48 | 86.36 | 77.26 | 82.85 | 77.03 |

Table 17: Task Accuracy(%) and ASR(%) under different injection rates(2% - 10%) and trigger size (2 - 20) in LAD-A dataset, target class 0, the attack mode is CAT+, the Original Accuracy is 88.54%.

| | 2% | | 5% | | 10% | |
|---|---|---|---|---|---|---|
| | ACC(%) | ASR(%) | ACC(%) | ASR(%) | ACC(%) | ASR(%) |
| 2 | 76.24 | 7.70 | 74.62 | 16.59 | 71.05 | 35.22 |
| 5 | 76.33 | 16.52 | 74.60 | 30.39 | 70.62 | 50.33 |
| 8 | 76.62 | 36.96 | 75.03 | 59.33 | 74.23 | 73.00 |
| 10 | 76.42 | 55.54 | 74.60 | 71.26 | 71.07 | 78.63 |
| 12 | 76.47 | 42.56 | 74.51 | 70.96 | 71.11 | 81.76 |
| 15 | 76.74 | 51.69 | 74.58 | 69.57 | 70.98 | 75.19 |

Table 18: Task Accuracy(%) and ASR(%) under different injection rates(2% - 10%) and trigger size (2 - 15) in LAD-E dataset, target class 0, the attack mode is CAT, the Original Accuracy is 77.82%.

(iii) Calculate each concept distance $\mathcal{Z}_{c_{select}}$ in selected concept:

$$\mathcal{Z}_{c_{select}} = \Sigma_{i=0}^{i=c_{num}} (c_i - c_{select})^2 \tag{27}$$

(iv) The Z-score for $c_a$ is defined as:

$$\mathcal{Z}(c_a) = \mathcal{Z}_{c_{select}} \mathcal{Z}(c_{select}, P_{select}) = \mathcal{Z}_{c_{select}} \left[ p^{(target|c_a)} - p_0 \right] / \left[ \frac{p_0(1 - p_0)}{p^{(target|c_a)}} \right] \tag{28}$$

## N  Complexity Analysis of CAT and CAT+

We provide a computational complexity analysis for our two trigger selection algorithms, CAT and CAT+. This analysis quantifies the one-off, offline cost required by an attacker to prepare the concept-based trigger. All experiments were conducted on the CUB dataset, and the trigger selection process for each setting was repeated five times to ensure reliable measurements. The results, including the mean, standard deviation, minimum, and maximum execution time in seconds, are presented in Table 23.

The results in Table 23 clearly delineate the trade-offs between the two methods and highlight the practical feasibility of both.

- **CAT is extremely efficient.** As a one-shot algorithm based on filtering and selecting from a pre-computed matrix, its execution time is consistently low, averaging around 1-2 seconds regardless of the trigger size. This makes it a highly practical and low-cost attack strategy.

- **CAT+ incurs a higher, yet entirely manageable, computational cost.** Its iterative, greedy search for an optimal concept set results in a runtime that scales approximately linearly with the desired trigger size. Even for a large trigger of 20 concepts, the selection process takes under 30 seconds on average.

|  | 2% | | 5% | | 10% | |
|---|---|---|---|---|---|---|
|  | ACC(%) | ASR(%) | ACC(%) | ASR(%) | ACC(%) | ASR(%) |
| 2 | 83.96 | 3.86 | 74.42 | 37.39 | 71.97 | 57.91 |
| 5 | 76.18 | 26.17 | 74.83 | 40.31 | 71.40 | 56.91 |
| 8 | 76.56 | 52.43 | 74.96 | 65.31 | 72.06 | 75.52 |
| 10 | 76.76 | 29.67 | 74.91 | 46.50 | 71.76 | 70.85 |
| 12 | 76.67 | 22.76 | 74.96 | 37.33 | 71.52 | 45.07 |
| 15 | 77.19 | 12.67 | 74.80 | 28.31 | 72.33 | 35.93 |

Table 19: Task Accuracy(%) and ASR(%) under different injection rates(2% - 10%) and trigger size (2 - 15) in LAD-E dataset, target class 0, the attack mode is CAT+, the Original Accuracy is 77.82%.

|  | 2% | | 5% | | 10% | |
|---|---|---|---|---|---|---|
|  | ACC(%) | ASR(%) | ACC(%) | ASR(%) | ACC(%) | ASR(%) |
| 2 | 87.86 | 23.95 | 85.40 | 33.74 | 82.72 | 47.41 |
| 5 | 87.95 | 75.89 | 85.42 | 77.59 | 82.58 | 84.56 |
| 8 | 87.40 | 94.61 | 85.62 | 97.22 | 82.84 | 96.69 |
| 10 | 87.90 | 90.04 | 85.81 | 93.80 | 82.60 | 93.82 |

Table 20: Task Accuracy(%) and ASR(%) under different injection rates(2% - 10%) and trigger size (2 - 10) in LAD-F dataset, target class 0, the attack mode is CAT, the Original Accuracy is 89.59%.

In conclusion, while CAT+ requires more computational effort during the attack's preparation phase, this cost is a modest, one-time investment. As demonstrated in our main results (e.g., in Section 7), this additional cost is overwhelmingly justified by the substantial increase in Attack Success Rate (ASR) it achieves (e.g., from 59.28% to 93.01% on CUB at a 10% injection rate). This makes CAT+ a highly potent and cost-effective strategy for a determined adversary.

## Ethics Statement

This work introduces and explores the concept of backdoor attacks in CBMs, a topic that inherently involves considerations of ethics and security in machine learning systems. The research aims to shed light on potential vulnerabilities in CBMs, with the intention of prompting further research into defensive strategies to protect against such attacks.

While our work demonstrates how CBMs can be compromised, we emphasize that the knowledge and techniques presented should be used responsibly to improve system security and not for malicious purposes. We acknowledge the potential risks associated with publishing methods for implementing backdoor attacks; however, we believe that exposing these vulnerabilities is a crucial step toward understanding and mitigating them.

Researchers and practitioners are encouraged to use the findings of this study to develop more robust and secure AI systems. It is our hope that by bringing attention to these vulnerabilities, we can collectively advance the field towards more transparent, interpretable, and secure machine learning models.

## Reproducibility Statement

To ensure the reproducibility of our results, we provide detailed descriptions of the datasets used, preprocessing steps, model architectures, and experimental settings within the paper and its appendices. Specifically:

**- Datasets:** We utilize the publicly available CUB and AwA datasets, with specific preprocessing steps outlined in Appendix H.

| | 2% | | 5% | | 10% | |
|---|---|---|---|---|---|---|
| | ACC(%) | ASR(%) | ACC(%) | ASR(%) | ACC(%) | ASR(%) |
| 2 | 83.96 | 3.86 | 82.18 | 7.33 | 79.59 | 13.69 |
| 5 | 83.65 | 11.23 | 81.45 | 22.14 | 78.06 | 32.13 |
| 8 | 83.30 | 55.32 | 81.20 | 67.65 | 78.69 | 81.24 |
| 10 | 83.91 | 64.13 | 82.87 | 79.68 | 79.40 | 80.14 |
| 12 | 83.96 | 75.48 | 82.42 | 83.59 | 78.71 | 90.57 |
| 15 | 83.98 | 72.30 | 81.63 | 81.59 | 79.14 | 85.43 |

Table 21: Task Accuracy(%) and ASR(%) under different injection rates(2% - 10%) and trigger size (2 - 15) in LAD-V dataset, target class 0, the attack mode is CAT+, the Original Accuracy is 84.30%.

| Model | Accuracy | Precision | Recall | F1 Score |
|---|---|---|---|---|
| Human-1 | 0.517 | 0.508 | 1.000 | 0.674 |
| Human-2 | 0.483 | 0.471 | 0.267 | 0.340 |
| Human-3 | 0.483 | 0.333 | 0.033 | 0.061 |
| GPT4v-1 | 0.433 | 0.464 | 0.867 | 0.605 |
| GPT4v-2 | 0.467 | 0.483 | 0.933 | 0.636 |
| GPT4v-3 | 0.483 | 0.492 | 0.967 | 0.652 |

Table 22: Classification Metrics Comparison

| Trigger | CAT | | | | CAT+ | | | |
|---|---|---|---|---|---|---|---|---|
| Size | Min(s) | Max(s) | Mean(s) | Std(s) | Min(s) | Max(s) | Mean(s) | Std(s) |
| 2 | 0.72 | 2.65 | 1.16 | 0.76 | 3.36 | 3.98 | 3.73 | 0.25 |
| 5 | 0.62 | 1.05 | 0.80 | 0.15 | 8.75 | 9.18 | 8.96 | 0.16 |
| 10 | 0.82 | 1.90 | 1.11 | 0.43 | 15.21 | 16.38 | 15.86 | 0.44 |
| 15 | 0.43 | 3.88 | 1.30 | 1.38 | 22.69 | 23.06 | 22.86 | 0.14 |
| 20 | 0.67 | 4.73 | 1.75 | 1.68 | 28.90 | 29.47 | 29.29 | 0.23 |

Table 23: Time complexity comparison for **trigger selection** in CAT and CAT+ under different trigger sizes. All experiments are conducted on the CUB dataset, and each setting is repeated five times to report the execution time (in seconds). The cost represents a one-off, offline effort for the attacker during the attack preparation phase.

**- Model Architecture:** Details about the Concept Bottleneck Model (CBM) architecture, including the use of a pretrained ResNet50 and modifications for each dataset, are provided in Section 7.

**- Experimental Settings:** The experimental setup, including training hyperparameters, batch sizes, learning rates, and data augmentation techniques, are thoroughly described in Section 7.

**- Attack Implementation:** The methodology for implementing our proposed CAT and CAT+ attacks, including concept selection, trigger embedding, and iterative poisoning strategies, is elaborated in Sections 4 and 5, with additional insights provided in the appendices.

Furthermore, to foster transparency and facilitate further research in this area, we commit to making our code publicly available upon publication of this paper. This includes scripts for preprocessing data, training models, executing backdoor attacks, and evaluating model performance and attack effectiveness.

