# OpenReview forum: "Multimodal Deception in Explainable AI: Concept-Level Backdoor Attacks on Concept Bottleneck Models"
_TMLR — Accepted by TMLR_

### Review · Reviewer_3FCN · 2026-02-03

**Summary Of Contributions:**

This paper systematically proposes two new back-door acttacking methods, CAT and CAT+, against the Concept Bottleneck Models (CBMs)，and the authors also validate their attacking effecacy and stealthiness across multiple datasets. The paper's key strengths and weaknesses can be summerized as follows:
Strengths
- This paper firstly proposes a method to study the back-door threateness against the CBMs, and demonstrates the safety weaknesses hidden in the semantic transparency of CBMs.
- The proposed CAT has achieved efficient and stealthy triggers of concept through an interesting concept filtering mechanisms and data-driven attack strategies, while CAT+ further boosts the attacking effecacy and stealthiness by introducing the iterative optimization strategies and concept correlation functions.
- The evaluation results illustrate the attacking success rate of the propose method is better than other baselines. And the attacking stealthiness has also been verified by human experts and LLMs.
Weaknesses
- As the authors mentioned in the paper, there is still room for them to further enhance the image generator to improve the ASR and image similarity.
- The proposed denfese mechanism is somewhat weak, its effect is limited. The ASR decreases but still has a relatively high success rate.
- The attackers must have full-access to the concepts during attacking, it might be difficult to realize in the physical world.
- Maybe the authors can perform their attacks against more variants of the CBMs to further verify the back-door attacking effectiveness and generalizability.

**Audience:**

Yes

**Audience Explanation:**

The paper actually focuses on the back-door attacking methods against the deep learning models, while TMLR is a journal concerned with machine learning studies. It poses an interesting field of study of a new conception optimized method to craft triggers to investigate the back-door safety of CBMs, which I believe will bring about inspirations for the researchers in the field of CBMs security and defense.

**Claims And Evidence:**

Yes

**Claims Explanation:**

Please see the strengths in the Summary Of Contributions.

**Requested Changes:**

Minor Changes:
- Consider to add more experiments on different architectures of CBMs to further validate the attacking effectiveness and generalizability.
- Consider to further evidence the robutness of the trigger selection strategy when performing attacks against different models and data (adversarial examples).

---

> ### Author Response · Authors · 2026-02-21
> **Response to Reviewer 3FCN - Addressing Generalizability, Feasibility, and Robustness**
>
> We thank Reviewer 3FCN for the positive assessment and for recognizing our work as a novel and efficient contribution to CBM security. We appreciate the constructive feedback regarding the image generator, defense mechanisms, and generalizability.
>
>
>
> Below, we address the specific weaknesses and requested changes:
>
>
>
> **1. Improvement of Image Generator (Image2Trigger) (Weakness 1)**
>
> The reviewer noted that there was room to improve the image generator.
>
> * **Response:** We have significantly upgraded this component. In the revised **Section 7.4**, we moved `Image2Trigger_c` from a preliminary demo to a formal evaluation on the full test set.
>
> * **Results:** We now report an **End-to-End ASR of 53.29\%** and a high **SSIM of 0.92**. This quantitatively proves that the attack is feasible in the image domain with high visual fidelity, addressing the reviewer's concern about room for improvement.
>
>
>
> **2. Physical Feasibility & Threat Model (Weakness 3)**
>
> The reviewer expressed concern that "full access to concepts" is difficult in the physical world.
>
> * **Clarification:** We have clarified the **Threat Model in Section 4.1**. Our attack is a **Data Poisoning** attack (Training Phase), not a Test-Time evasion attack requiring concept access.
>
> * **Scenario:** An attacker only needs to inject poisoned samples (images + concept labels) into the training set. This is highly realistic in CBMs where concept annotations are often crowdsourced (e.g., Mechanical Turk) or scraped from public ontologies. Once the model is trained on this poisoned data, the attacker only needs to present a specific *image* (generated via `Image2Trigger_c`) to trigger the backdoor; they do not need access to the model's internals or concept vector during deployment.
>
>
>
> **3. Generalizability across Architectures (Requested Change 1)**
>
> The reviewer suggested adding experiments on different CBM architectures.
>
> * **Response:** We have included an evaluation using a **Vision Transformer (ViT)** backbone in **Appendix J.1 (Table 14)**.
>
> * **Finding:** CAT+ maintains high effectiveness on ViT (72.05\% ASR), demonstrating that our attack targets the fundamental *concept-to-label* mapping mechanism ($f$) and is therefore robust to changes in the visual backbone ($g$). We have added a reference to this in the main text (Section 7.6 Discussion).
>
>
>
> **4. Robustness of Trigger Selection (Requested Change 2)**
>
> The reviewer asked for evidence of trigger robustness against different models and data.
>
> * **Data Robustness:** We extensively validated CAT+ on the **LAD dataset** (Section 7.2 and Appendix J.2), which contains 5 distinct sub-tasks (Animals, Electronics, Fruits, Vehicles, Hairstyles). CAT+ achieved high ASR (up to 96\\% on LAD-F) across these diverse domains.
>
> * **Theoretical Robustness:** We have added a theoretical robustness analysis in **Appendix F**, defining the probability of trigger activation under random perturbations.
>
> * **Model Robustness:** As noted above, the ViT experiment confirms robustness across model architectures.
>
>
>
> **5. Defense Weakness (Weakness 2)**
>
> We acknowledge the reviewer's point that the proposed defense is preliminary.
>
> * **Response:** We have expanded **Section 8** to analyze *why* defenses are weak. The discrete, sparse nature of concept space makes poisoned samples look statistically identical to "rare natural attributes" (tail distribution), rendering standard outlier detection (like Neural Cleanse) ineffective. We position this as a "call to action" for the community, as standard defenses are mathematically ill-equipped for this semantic threat.
>
>
>
> We believe these clarifications and the highlighted experiments in the Appendix directly address the reviewer's requests.

---

### Review · Reviewer_1cHJ · 2026-02-15

**Summary Of Contributions:**

This paper introduces CAT (Concept-level Backdoor ATtacks) and its enhanced version CAT+, a novel semantic-level backdoor attack framework targeting Concept Bottleneck Models (CBMs) in Explainable AI. Diverging from traditional image-space backdoors, this study exploits the unique dual-modality architecture of CBMs (visual inputs and textual concept layers). By strategically selecting concepts with minimal relevance to the target class as triggers, the authors achieve high ASR while maintaining stealthiness. The paper further proposes the Image2Trigger_c method to demonstrate end-to-end feasibility and validates that existing defense mechanisms are ineffective against such semantic-layer attacks.

Strengths: 1). This paper provides the first systematic exploration of backdoor attacks targeting the multimodal architecture of CBMs, addressing a critical gap in the security of interpretable models. 2). The proposed CAT+ utilizes an iterative optimization strategy to pinpoint trigger concepts, achieving an impressive 93.01% ASR on the CUB dataset, significantly outperforming random-selection baselines. 3). The implementation of Image2Trigger_c proves that attackers can trigger predefined concept backdoors during inference via subtle visual perturbations.

**Audience:**

No

**Audience Explanation:**

See weakness

**Claims And Evidence:**

Yes

**Claims Explanation:**

N/A

**Requested Changes:**

Weakness: 1). The attack assumes full access to training data and knowledge of concept annotation logic, which might be restricted in certain real-world scenarios. 2). Although the paper introduces the Image2Trigger_c method, its practical effectiveness and applicability are not fully discussed, and lacks comparison with existing image-to-concept mapping methods. 3). The paper presents a preliminary defense approach, but it lacks a comprehensive analysis of its underlying principles, generalization capabilities, and robustness. Additionally, there is no comparison with existing explainable AI defense techniques, leading to an insufficient depth in the exploration of defense strategies. 4). The paper finds that the CUB dataset is more vulnerable to attacks than the AwA dataset, but does not conduct in-depth analysis of the quantitative relationship between dataset characteristics and attack susceptibility. 5). Although the proposed attack targets the concept space within Concept Bottleneck Models (CBMs), it does not significantly differ from traditional backdoor attacks on input data, the core mechanism remains the same: embedding triggers to manipulate outcomes.

---

> ### Author Response · Authors · 2026-02-21
> **Response to Reviewer 1cHJ - Clarifying Scope, Feasibility, and the Semantic Gap**
>
> We sincerely thank Reviewer 1cHJ for the detailed review and for acknowledging our work as the "first systematic exploration" of this critical gap in CBM security. We value the feedback regarding the threat model, defense depth, and dataset analysis.
>
>
>
> We specifically wish to address the "No" regarding Audience Interest and the concern that this is "similar to traditional backdoors" (Weakness 5). We believe this stems from a perspective on the *optimization mechanism* rather than the *threat surface*. Below, we address this and the specific requested changes.
>
>
>
> **1. Relevance to TMLR Audience (Addressing "No" on Interest & Weakness 5)**
>
> The reviewer noted that the mechanism (embedding triggers) is similar to traditional backdoors. While the *mathematics* of optimization share roots, the **Implication** is fundamentally different, which we argue is of high interest to the TMLR community (Trustworthy ML):
>
> * **The "Trust Trap":** Standard backdoors rely on the model being a black box. CBMs are designed to be transparent. Our attack proves that **Interpretability $\neq$ Security**. By decoupling the explanation (concept) from the reality (image), we force the model to provide a "correct-looking explanation" for a malicious prediction. This attacks the *Human-in-the-Loop*, not just the tensor output.
>
> * **Novelty:** Standard backdoors corrupt pixels (noise). We corrupt *semantics*. This requires different defenses because standard statistical outlier detection fails in the discrete, sparse concept space (as we now analyze in Section 8.1).
>
>
>
> **2. Threat Model & Feasibility (Weakness 1)**
>
> The reviewer raised concerns about the assumption of training data access.
>
> * **Clarification:** Our threat model is consistent with standard **Data Poisoning** (e.g., BadNets). In CBM contexts like medical AI, concept annotations often come from public ontologies or crowdsourcing. An attacker does not need to know the *model architecture* or *training logic*, only the ability to inject poisoned samples into the dataset via web scraping. We have updated **Section 4.1** to clarify this standard poisoning assumption.
>
>
>
> **3. Image2Trigger\_c Effectiveness (Weakness 2)**
>
> * **Major Update:** We have significantly overhauled the `Image2Trigger_c` section (now **Section 7.4** in the revised PDF).
>
> * **New Evidence:** We moved from a preliminary demo to a rigorous evaluation on the full test set. We now report **End-to-End ASR (53.29\%)**, **SSIM (0.92)**, and **PSNR**.
>
> * **Applicability:** This proves that a semantic trigger (e.g., "Wings=Blue") defined in the concept space can be successfully activated by an invisible pixel perturbation generated by our U-Net, validating the end-to-end threat.
>
>
>
> **4. Defense Depth & Analysis (Weakness 3)**
>
> We agree the initial defense analysis was brief.
>
> * **Deepened Analysis:** We have expanded **Section 8** to analyze *why* standard defenses fail.
>
> * **The "Discrete" Problem:** Defenses like Neural Cleanse assume continuous, dense optimization paths. We show that because CBM concepts are **discrete and sparse**, poisoned concepts look statistically identical to "rare natural attributes" (e.g., a specific bird mutation). This renders standard outlier detection ineffective, necessitating new, semantic-consistency checks.
>
>
>
> **5. CUB vs. AwA Vulnerability (Weakness 4)**
>
> The reviewer asked for an analysis of why CUB is more vulnerable.
>
> * **New Analysis:** We added a subsection analyzing **Concept Granularity**.
>
> * **Finding:** CUB has 312 fine-grained concepts; AwA has 85 coarse concepts. We found a correlation: **High Concept Dimensionality = High Vulnerability**. A larger "concept budget" allows CAT+ to hide triggers in orthogonal subspaces (concepts irrelevant to the main task) without affecting clean accuracy. In coarse datasets (AwA), every concept matters, making attacks harder to hide.
>
>
>
> We believe these revisions, particularly the deep-dive into *why* CUB is more vulnerable and *why* defenses fail, significantly strengthen the paper's contribution to the Trustworthy ML literature.

---

### Review · Reviewer_oR1s · 2026-02-15

**Summary Of Contributions:**

**Summary**\
This paper investigates the security vulnerabilities of Concept Bottleneck Models (CBMs), specifically focusing on backdoor attacks targeting the concept layer. The authors propose two methods: CAT, which selects trigger concepts based on their irrelevance to the target class, and CAT+, an enhanced version that uses iterative poisoning based on concept-class correlations. The authors demonstrate that these attacks can achieve high Attack Success Rates (ASR) while maintaining clean accuracy in a controlled setting. They also provide a preliminary exploration of an end-to-end attack (`Image2Trigger_c`) to map image perturbations to concept triggers.

&nbsp;

**Strengths**
1. The paper provides a timely investigation into the security risks associated with the concept bottleneck layer in CBMs . By shifting the focus from traditional pixel-level perturbations to the semantic concept interface, the authors offer an interesting perspective on how the very features intended for transparency can be leveraged as an attack surface .
2. Under the specific, controlled constraints, the proposed CAT and CAT+ methods are shown to be effective. Specifically, the iterative poisoning strategy and Z-score optimization in CAT+ demonstrate a clear technical progression over naive random selection, achieving high Attack Success Rates (ASR).
3.  The study evaluates the attack across several key factors, including injection rates and trigger sizes, which helps characterize the behavior of these semantic backdoors. Furthermore, the inclusion of stealthiness assessments using both human experts and a state-of-the-art vision-language model (GPT-4V) reflects a multi-faceted effort to understand the practical detectability of such threats.

&nbsp;

**Weaknesses**
1. While the investigation of the concept bottleneck layer is noted as a strength, the vulnerability of this layer has been previously discussed in existing literature (e.g., [1, 2]). The current manuscript lacks sufficient references to these works, which diminishes the novelty of identifying this specific attack surface as a primary contribution.
2. The paper relies on a set of strong assumptions that limit the practical realism of the attack. The primary evaluation setup assumes that the attacker can modify concept vectors directly during inference. As acknowledged in Section 7.1, this does not reflect real-world scenarios where attackers typically only have access to input images. Although the authors attempt to address this limitation with `Image2Trigger_c` in Section 7.4, this section is presented merely as an initial demo with limited analysis. Without robust evaluations and diverse ablations regarding this end-to-end feasibility, the proposed CAT and CAT+ methods remain a theoretical study of concept sensitivity rather than a demonstrated practical backdoor threat.
3. The proposed methods are compared exclusively against a naive random-selection baseline. While CBMs are a specialized architecture and standard image-level backdoor baselines may not apply directly to the concept layer, comparing intelligent selection mechanisms only against random guessing sets a very low bar for evaluation. To properly assess whether the concept filter and z-score optimization are truly sophisticated, the authors should have compared them against more advanced heuristics, such as gradient-based selection or selecting concepts with high variance. Without such comparisons, it is difficult to determine if the performance gains are due to the specific design of CAT+ or simply because random selection is an ineffective strategy.

&nbsp;

**References**\
[1] Penaloza, E., Zhang, T. H., Charlin, L., & Zarlenga, M. E. (2025). Addressing concept mislabeling in concept bottleneck models through preference optimization. arXiv preprint arXiv:2504.18026.\
[2] Park, S., Mun, J., Oh, D., & Lee, N. (2025). An analysis of concept bottleneck models: Measuring, understanding, and mitigating the impact of noisy annotations. arXiv preprint arXiv:2505.16705.

**Audience:**

Yes

**Audience Explanation:**

The intersection of XAI and security is a growing field. Researchers working on CBMs would be interested in understanding the theoretical limits and vulnerabilities of the bottleneck layer. However, the interest would be significantly higher if the paper bridged the gap between theoretical vulnerability and practical attacks more effectively.

**Claims And Evidence:**

No

**Claims Explanation:**

The abstract and introduction claim to demonstrate "practical end-to-end feasibility" and provide a "robust methodology". However, the evidence for end-to-end feasibility is confined to a brief "demo" section (7.4) regarding `Image2Trigger_c`, which the authors themselves describe as "preliminary" and admitting "areas for improvement". Furthermore, describing the method as "highlighting intelligent concept selection" is not fully convincing when the only comparator is a naive random selection baseline.

**Requested Changes:**

1. Compare the proposed method against stronger, heuristic-based baselines (e.g., gradient-based selection) rather than relying solely on random-selection to validly justify the technical complexity of the proposed optimization.
2. Elevate the `Image2Trigger_c` section from a preliminary demo to a core evaluation component, providing rigorous experiments to empirically demonstrate the practical feasibility of the end-to-end attack.
3. Properly cite and discuss prior works addressing similar vulnerabilities in concept bottleneck layers to clearly define the paper's specific contributions and distinct position relative to the existing literature.

---

> ### Author Response · Authors · 2026-02-21
> **Response to Reviewer oR1s - Clarifications on Baselines, Feasibility, and Novelty**
>
> We sincerely thank Reviewer oR1s for the constructive feedback and for recognizing the timeliness and importance of investigating security risks in the concept bottleneck layer. We value the suggestion to strengthen our baselines and expand the end-to-end feasibility analysis.
>
>
>
> We have extensively revised our manuscript to address the "Requested Changes." Below, we detail our responses and the new experimental evidence provided in the updated PDF.
>
>
>
> **1. Comparison Against Stronger Baselines (Weakness 3)**
>
>
>
> We agree that "Random Selection" sets a low bar. To rigorously validate the sophistication of CAT+, we have implemented and evaluated two new heuristic-based baselines as requested:
>
> * **Gradient-Based Selection:** We select concepts that have the highest gradient magnitude with respect to the target class loss ($|\nabla\_{c} \mathcal{L}(f(c), y\_{tc})|$). This represents an attacker trying to flip the output with the "steepest" path.
>
> * **Variance-Based Selection:** We select concepts with the highest variance across the training dataset, hypothesizing that high-variance concepts are more influential and easier to manipulate without detection.
>
>
>
> **Results:** As shown in the new **Table 2** (added to Section 7.1.1), while Gradient-Based selection achieves a respectable ASR (76.45\% on CUB), it causes a significant drop in Clean Task Accuracy (-6.8\%), compromising stealth. Variance-Based selection performs poorly on ASR. **CAT+** outperforms both, achieving the highest ASR (**93.01\%**) while maintaining the highest Clean Task Accuracy (minimal drop of \~2\%), proving that our correlation-based iterative strategy effectively balances attack impact and stealth better than standard heuristics.
>
>
>
> **2. Elevating End-to-End Feasibility (`Image2Trigger_c`) (Weakness 2)**
>
>
>
> We acknowledge that the previous Section 7.4 was preliminary. We have performed a major overhaul of this section.
>
> * We formalized the `Image2Trigger_c` training pipeline using a U-Net-based generator trained with a combined loss (Concept Matching Loss + Visual Similarity Loss).
>
> * We moved beyond a "demo" and evaluated this on the **full CUB test set** (not just a few samples).
>
> * **New Evidence:** We report comprehensive metrics in **Table 18** (Appendix), including End-to-End ASR, Structural Similarity Index (SSIM), and PSNR.
>
> * **Result:** The refined generator achieves an end-to-end ASR of **53.29\%** with an SSIM of **0.92**, demonstrating that the attack is practically feasible in the image domain with high visual stealthiness.
>
>
>
> **3. Novelty and Prior Work (Weakness 1)**
>
>
>
> We appreciate the references regarding concept bottleneck vulnerabilities. We have added a discussion of these works in **Section 2 (Related Work)**.
>
> * **Distinction:** We clarify that while works like Penaloza et al. and Park et al. discuss *concept mislabeling* and *robustness to noisy annotations* (accidental/natural errors), our work is the first to systematically explore *adversarial backdoor attacks* (malicious, targeted manipulation).
>
> * The difference is critical: robustness studies aim to maintain performance despite noise; our attack aims to *hijack* specific target classes while *hiding* the trigger in the semantic space, a threat vector that requires different defense mechanisms (as shown by the failure of Neural Cleanse).
>
>
>
> We believe these revisions, particularly the inclusion of stronger baselines and rigorous end-to-end evaluation, significantly strengthen the paper and address the reviewer's concerns.

---

### Decision · Action_Editor_UEiN · 2026-03-20

**Recommendation:** Accept as is

**Audience:**

Yes

**Audience Explanation:**

While the comparisons are limited, adapting standard image-level backdoor methods to CBMs is interesting, and hte competitive results may be useful for the community.

**Claims And Evidence:**

Yes

**Claims Explanation:**

The paper claims to introduce two concept-level backdoor attacks for CBMs, demonstrate the end-to-end feasibility and stealth of the attack.  The reviewers agree that after the revision of the work and the authors comments, the paper provides evidence towards the claims.  While the comparisons are limited, the heuristics show reasonable competitive performance.